# PTPN2 copper-sensing relays copper level fluctuations into EGFR/CREB activation and associated *CTR1* transcriptional repression

Matthew O. Ross [1]✉, Yuan Xie [2], Ryan C. Owyang [1], Chang Ye [1], Olivia N. P. Zbihley [2], Ruitu Lyu [1], Tong Wu [1], Pingluan Wang[1], Olga Karginova[3], Olufunmilayo I. Olopade [3], Minglei Zhao [2] & Chuan He [1,2,4]✉

Fluxes in human copper levels recently garnered attention for roles in cellular signaling, including affecting levels of the signaling molecule cyclic adenosine monophosphate. We herein apply an unbiased temporal evaluation of the signaling and whole genome transcriptional activities modulated by copper level fluctuations to identify potential copper sensor proteins responsible for driving these activities. We find that fluctuations in physiologically relevant copper levels modulate EGFR signal transduction and activation of the transcription factor CREB. Both intracellular and extracellular assays support $Cu^{1+}$ inhibition of the EGFR phosphatase PTPN2 (and potentially PTPN1)–via ligation to the PTPN2 active site cysteine side chain–as the underlying mechanism. We additionally show *i*) copper supplementation drives weak transcriptional repression of the copper importer *CTR1* and *ii*) CREB activity is inversely correlated with *CTR1* expression. In summary, our study reveals PTPN2 as a physiological copper sensor and defines a regulatory mechanism linking feedback control of copper stimulated EGFR/CREB signaling and *CTR1* expression.

Copper is an essential nutrient in the human diet with prototypical roles as metalloenzyme cofactors; however, excess copper induces cellular (redox) stress via formation of reactive oxygen/nitrogen species. Relatedly, the biochemical mechanisms regulating intracellular copper levels have been extensively characterized. There are two mammalian copper importers; the cuprous importer CTR1 (encoded by the *SLC31A1* gene)–responsible for the vast majority of copper internalization–and the cupric importer DMT1 (encoded by the *SLC11A2* gene). For expediency, we hereafter refer to the *SLC31A1* and *SLC11A2* genes and transcripts as *CTR1* and *DMT1*, respectively. Following reduction to $Cu^{1+}$ and import, copper is shuttled inside the cell by chaperone proteins (e.g., ATOX1) to subcellular locations for metallocofactor assembly or export.

Copper influx and efflux stimulate a wide range of human cellular responses, including cuproplasia (copper dependent cell growth and proliferation)[1], cuproptosis[1,2], neuronal activities[3–5], cell differentiation[6], and the epithelial to mesenchymal transition in cancer[7], as well as signaling pathways such as cyclic adenosine monophosphate (cAMP) signaling[8], MAPK/ERK, JAK-STAT, PI3K/Akt, NF-κB, and the NOD-like receptor protein 3 (NLRP3) inflammasome[9]. Furthermore, copper dyshomeostasis causes human diseases such as Menkes syndrome and Wilson disease, and elevated copper levels induce numerous pro-cancer activities (e.g., angiogenesis, metastasis, and cell proliferation)[10]. Relatedly, serum and tissue copper levels correlate with carcinogenesis and prognosis across diverse cancers[11]. Targeting copper homeostasis pathways has been proposed as

[1]Department of Chemistry, University of Chicago, Chicago, IL, USA. [2]Department of Biochemistry and Molecular Biology, University of Chicago, Chicago, IL, USA. [3]Department of Medicine, Center for Clinical Cancer Genetics and Global Health, University of Chicago, Chicago, IL, USA. [4]Howard Hughes Medical Institute, University of Chicago, Chicago, IL, USA. ✉e-mail: matthewross@uchicago.edu; chuanhe@uchicago.edu

promising therapeutic approaches against diverse diseases, including cancer[12,13].

Across the tree of life, copper stimulates cellular signaling/processes through either indirect mechanisms (change in cellular redox status, generic metal dyshomeostasis response, etc.) or direct binding to proteins. Common bacterial and yeast examples of the latter include copper sensing transcription factors (TFs), which alter copper homeostasis protein expression levels to regulate intracellular copper concentrations. In humans, it was recently discovered that copper binding to the enzyme PDE3B inhibits its cAMP degrading function, leading to elevated levels of the signaling molecule cAMP[8,14]. However, considerably less is known about the transcriptional programs altered by copper levels in humans; whether mammalian intracellular copper levels are even regulated by differential transcriptional/translational expression of copper homeostasis proteins has been an ongoing debate[15–17].

In this report, we apply an unbiased temporal evaluation of the whole genome transcriptional activities stimulated by fluctuations in copper levels under normal cellular growth conditions. We first employ the recently developed kethoxal assisted single stranded DNA (ssDNA) sequencing (KAS-seq) technique[18], which reports genomic ssDNA regions (corresponding to active transcription), to evaluate the transcriptional programs immediately activated following copper supplementation. By pairing these results with phosphorylation arrays to define copper stimulated signal transduction pathways, as well as transcriptomic, molecular biological, and biochemical investigations, we produce strong evidence that low level copper supplementation causes Cu$^{1+}$ binding to–and inactivation of–the protein tyrosine phosphatase (PTP) PTPN2 at the active site cysteine to drive activation of EGFR signal transduction, leading to MAPK/ERK/CREB (cAMP response element binding protein) activation. Moreover, we show elevated copper levels, whether supplemented into the extracellular media or via direct intracellular release using a copper ionophore, drive weak *CTR1* repression, and that CREB activity is inversely correlated with *CTR1* expression. We propose this pathway likely serves as a

feedback response to balance copper uptake and attenuate copper stimulated cAMP signaling, working in concert with other established mechanisms like copper stimulated CTR1 protein degradation[19]. Thus, copper /CREB stimulated *CTR1* transcriptional repression may drive the previously reported ~70% decrease in *CTR1* transcript levels following sustained activation of CREB and intracellular copper accumulation during neuronal differentiation[6,20–23].

## Results

### Rapid copper stimulated transcriptional/signaling responses

KAS-seq provides a genomic method that enables temporal profiling of the whole genome transcriptional activity to identify early responsive genes upon cellular stimulation or signaling. To study copper-induced signaling we applied KAS-seq to evaluate the genomic responses of A549 cells (non-small cell lung adenocarcinoma) to 10 μM CuCl$_2$ (low Cu) or 20–30 μM CuCl$_2$ (high Cu) supplemented growth media for 15 min, 2 h, and 4 h, relative to unsupplemented cells (Fig. 1a). For reference, healthy serum copper levels are ~10–30 μM (of which ~25% is readily exchangeable[24]), while up to ~3-fold increases have been observed in cancer patients[11,25]. Through KAS-seq we captured the sequence of copper stimulated transcriptional responses (Fig. 1b, Supplementary Figs. 1–2, Supplementary Data 1), including the immediately activated genes first induced by copper stimulation. Examining the gene body differential ssDNA levels revealed three genes significantly activated by both low and high copper incubation conditions: *EGR1* and *NR4A1* (15 min post-copper) as well as *CXCL2* (2 h post-copper), all of which are established CREB1 target genes[26] and are induced by the MAPK/ERK signal transduction pathway[27].

The implied MAPK/ERK pathway activation by copper led us to evaluate changes in phosphorylation and total protein levels among common signal transduction proteins via proteome profiler phosphokinase antibody arrays. Because A549 cells express mutant hyperactive KRAS and overexpress EGFR, which could obfuscate detection of copper stimulated MAPK/ERK activation, we selected a non-cancerous cell line for further validation and investigation. HEK 293 T cells were briefly

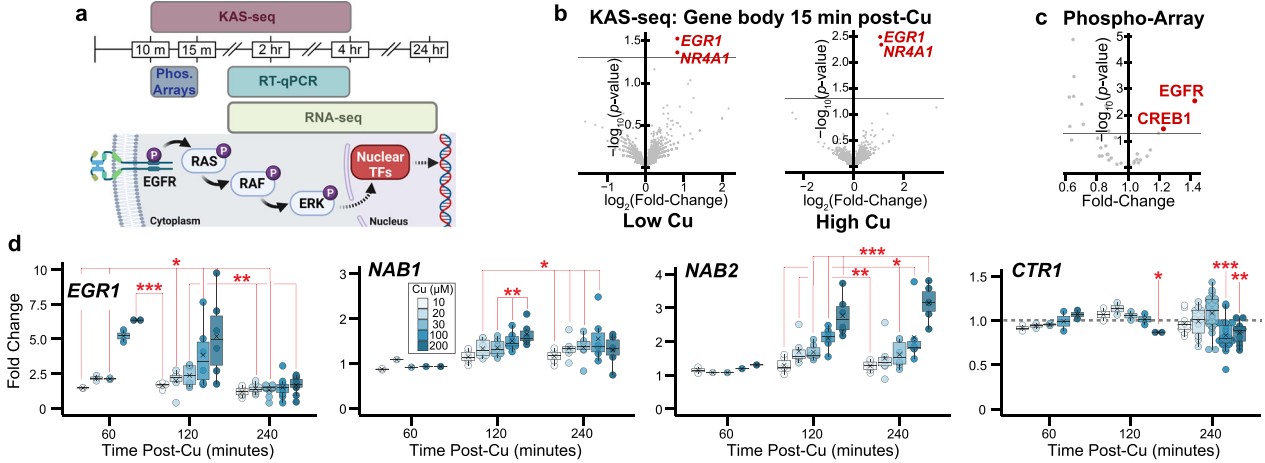

**Fig. 1 | EGFR/MAPK/ERK signaling and associated gene regulation. a** Schematic of the various techniques applied in this study to evaluate copper stimulated transcriptional/genomic dynamics, with the canonical EGFR/MAPK/ERK signaling pathway. Figure 1/panel a created with BioRender.com released under a Creative Commons Attribution-NonCommercial-NoDerivs 4.0 International license. **b** A549 KAS-seq gene body volcano plots (Cu-supplemented vs. unsupplemented) 15 min post-cell treatment. Significantly upregulated genes in red; horizontal gray lines denote *p* = 0.05; *n* = 2 biological replicates for each condition. Low Cu and High Cu conditions correspond to 10 and 20 μM CuCl$_2$ supplementation, respectively. *p*-value denotes two-sided Wald test as calculated by DESeq2, without correction for multiple observations. **c** Changes in HEK 293 T phosphorylation and/or total protein levels from proteome profiler phospho-kinase antibody arrays following

10 min, 20 μM CuCl$_2$ treatment (vs. untreated). *n* = 3 biological replicates for each condition. *p*-value denotes two-sided *t*-test without correction for multiple observations. **d** RNA-qPCR fold-changes of select transcripts following copper supplementation for indicated times into A549 cells. Box plot values defined as follows: box boundaries (median of first quartile to median of third quartile) and central horizontal line (median), mean expression value shown on the plot as an "x", whisker boundaries define 1.5 times the interquartile range. Data points correspond to biological replicates, *p*-values calculated as paired two-tailed *t*-test. Sample sizes were as follows for each transcript at 60 min, 120 min, and 240 min post-Cu; *EGR1* (*n* = 2, 6, 10), *NAB1* (*n* = 2, 6, 6), *NAB2* (*n* = 2, 6, 6), *CTR1* (*n* = 2, 2, 10–22). Source data are provided as a Source Data file, as are exact *p*-values. * denotes *p* < 0.05, ** denotes *p* < 0.01, and *** denotes *p* < 0.001.

(3 h) serum starved to attenuate serum-responsive signaling, then were supplemented with 20 μM $CuCl_2$. 10 min post-copper supplementation, cells were lysed and evaluated via the array relative to unsupplemented cell lysates (Supplementary Fig. 3). AKT phosphorylation was strongly, significantly diminished in response to copper treatment—interestingly, diminished AKT signaling was also previously detected when glial cells were treated with a cell permeable small molecule copper chelate[28]. There were two significantly activated protein phosphorylation sites: EGFR Y-1086 and CREB1 S-133 (Fig. 1c).

Phosphorylation of EGFR Y-1086 (as well as Y-1068) activates MAPK/ERK signaling[29], which subsequently drives phosphorylation of the nuclear TF CREB1 S-133 site[30] (among other TFs). The KAS-seq and phosphorylation array results collectively indicate that low level copper supplementation drives EGFR activation to initiate copper stimulated MAPK/ERK signaling, followed by downstream CREB1 activation and transcriptional responses.

## Copper drives EGFR, CREB activation and weak *CTR1* repression

To corroborate the transcriptional activities reported by KAS-seq and provide more quantitative assessment of this potential copper induced MAPK/ERK signal transduction, we performed RNA-qPCR to analyze copper homeostasis (*ATOX1* and *CTR1*) and MAPK/ERK induced (*EGR1*, *NAB1*, and *NAB2*) transcript expression levels from cells supplemented with 0–200 μM $CuCl_2$. All RNA-qPCR transcript expression levels are reported as fold-change relative to cells in media without $CuCl_2$ supplementation. As expected, *ATOX1* expression levels were unresponsive to copper supplementation (Supplementary Fig. 4). We measured a dose dependent increase in *EGR1* expression 1 h post-copper supplementation, followed by increased *NAB1* and *NAB2* expression 1–3 h later (Fig. 1d). *NAB1/2* expression is known to be strongly induced 1 h after MAPK/ERK induced *EGR1* expression[31,32]. *CTR1* was weakly repressed 4 h after ≥100 μM copper supplementation, consistent with prior RNase protection experiments[17]. We subsequently performed RNA-seq and confirmed that these copper stimulated changes in *NAB1*, *NAB2*, and *CTR1* expression levels were again detected 4 h post-copper supplementation (Supplementary Fig. 5). Pre-treatment of A549 cells with EGFR-inhibitors (2 μM gefitinib or 80 μg/mL cetuximab) followed by copper supplementation and RNA-qPCR validated copper stimulation of EGFR signaling as the driver of increased *EGR1* expression and *CTR1* repression (Supplementary Fig. 6). Knockdown or overexpression of *DMT1* had no significant effect on copper stimulated *EGR1* expression activation (Supplementary Fig. 7).

Our observation of copper stimulated EGFR activation suggested that MAPK/ERK downstream TFs may be responsible for *CTR1* transcriptional repression. As our results defined copper supplementation drives CREB phosphorylation—and given the aforementioned role of copper in cAMP signaling—we evaluated CREB as the candidate copper stimulated *CTR1* repressor TF. We analyzed previous chromatin immunoprecipitation-sequencing (ChIP-seq) experiments, and found significant CREB1 binding to the *CTR1* promoter in the majority of human cell lines in the ChIP-Atlas as well as in the CREB Target Gene Database[33–35] (Fig. 2a, Supplementary Fig. 8, Supplementary Table 1). The *CTR1* promoter *CRE* (cAMP response element) does not contain a proximal TATA box (Supplementary Table 1), the presence of which is generally associated with CREB induced gene expression[36], consistent with a role of CREB induced *CTR1* repression.

We then sought direct experimental evidence of CREB induced *CTR1* repression. We first evaluated *CTR1* levels in A549 cells via RNA-qPCR following *CREB1*-siRNA transfection (Fig. 2b, knockdown validation provided in Supplementary Fig. 9); *CREB1* knockdown significantly increased *CTR1* expression. Notably, the same results were also observed in a *CREB1* knockdown RNA-seq study in K562 (leukemia) cells (Supplementary Table 2)[37]. We next assessed chemical inhibition of CREB1 via supplementation of the small molecule CREB inhibitor 666-15, and similarly found *CTR1* levels were significantly

increased (Fig. 2b). By contrast, we found cells transfected with a pCMV vector driving constitutive *CREB* expression significantly (albeit weakly) repressed *CTR1* levels (Fig. 2b). Collectively, these results revealed CREB as the downstream copper stimulated repressor TF of *CTR1*.

## NGS repositories support a link between CREB and *CTR1* levels

To demonstrate physiological relevance of this pathway, we next sought in vivo evidence of *CTR1* repression via CREB. We plotted mean ChIP-signal enrichments from mouse tissue and blood CREB1 ChIP-seq experiments in the ChIP-Atlas (Fig. 2a, Supplementary Fig. 8). These ChIP-seq results confirmed binding of CREB1 at the *Ctr1* promoter inside living mammals. Relatedly, a prior study found mice injected with FGF23 (a MAPK/ERK activator kidney hormone) exhibited pronounced renal repression of *Ctr1*[38].

We next evaluated correlations between *CTR1* levels with expression levels of MAPK/ERK responsive genes (*EGR1*, *NAB1*, and *NAB2*) and the CREB1 target gene *NR4A1* across the five vital organs in healthy human tissues using Genotype-Tissue Expression (GTEx) RNA-seq datasets. We assessed whether activation of the MAPK/ERK/CREB signaling, as inferred by increased *EGR1*, *NAB1*, *NAB2*, and *NR4A1* expression levels, is associated with repression of *CTR1*. In all tissues, transcript correlations with *CTR1* were negative or statistically insignificant (Fig. 2c, Supplementary Table 3). As *EGR1*, *NAB2*, and *NR4A1* expression tend to be transient and stimulus induced[39,40], the observed inter-tissue conserved negative correlations support stimulus responsive *CTR1* repression by CREB1 in vivo. Note that *CREB1* levels are also negatively correlated with *CTR1* levels across the aforementioned tissues (excluding the non-significant correlation in the heart tissue, Supplementary Fig. 10).

For further in vivo evidence, and given the important roles of EGFR/MAPK/ERK/CREB in lung adenocarcinoma, we evaluated the Cancer Genome Atlas (TCGA) lung adenocarcinoma RNA-seq results, comparing MAPK/ERK activating $KRAS^{G12X}$ mutant (X is A, C, D, S, or V) with non-variant samples. As expected, MAPK/ERK stimulated and CREB1 target gene transcripts were significantly increased while *CTR1* levels were significantly (again, albeit weakly) decreased in $KRAS^{G12X}$ mutant relative to non-variant samples (Fig. 2d). Collectively, our results provide strong evidence that *CTR1* expression levels are repressed by CREB activation.

## Whole transcriptome profiling of copper stimulated activities

To investigate this copper induced regulatory pathway, as well as demonstrate reversibility of the copper stimulated CREB1 TF activation response, we next evaluated differences in gene expression across the transcriptome induced by copper depletion (via supplementation of tetrathiomolybdate (TM)) or copper supplementation using RNA-seq. Note: TM was chosen as a copper chelator (as opposed to $Cu^{1+}$/ $Cu^{2+}$ specific, or intra-/extracellular specific) as a way to sequester both extra- and intracellular copper. We found a close correspondence between transcriptomic responses 2 h post-copper supplementation and 24 h post-TM supplementation (Fig. 3a, b, Supplementary Data 2–3). Pathway and process enrichment analysis of differentially expressed transcripts (FDR-q < 0.05) post-treatment corroborated known copper stimulated pathways, supporting that these responses are regulated at the transcriptional level. Four ontologies of particular relevance were enriched in both the TM and copper supplemented differentially expressed genes (that is, transcripts significantly up- or downregulated, Fig. 3c,d, Supplementary Data 4–5): (i) regulation of (MAP) kinase activity, (ii) response to growth factor (including TGF-β— an EGFR transactivator[41]—response ontologies), as well as the related (iii) epithelial cell differentiation and (iv) cell/tube morphogenesis. Moreover, consistent with the known role of copper in stimulating cellular differentiation, evaluation of transcripts significantly repressed in response to TM also revealed enrichment of terms (iii) and (iv)

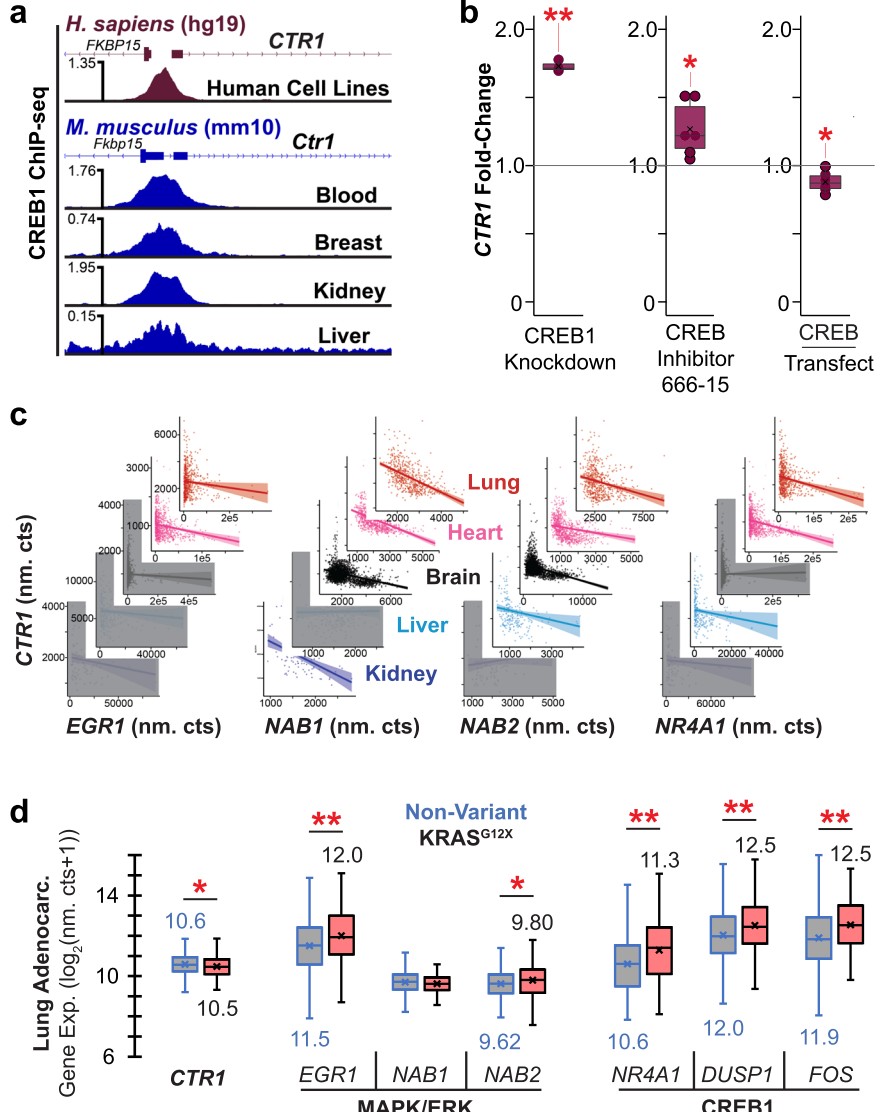

**Fig. 2 | Transcriptomic correlations of MAPK/ERK/CREB1 activation and *CTR1* repression. a** Mean enrichment of CREB1 ChIP-seq experiments generated from ChIP-Atlas *Homo sapiens* cell lines (*Top*) and *Mus musculus* tissues and blood (*Bottom*). Individual ChIP-seq experiments presented in SI Appendix Fig. S7. **b** A549 *CTR1* transcript level box plots following modulation of CREB activity levels. (*Left*) 24 h post-CREB1-knockdown, relative to cells transfected with negative control (non-targeting) siRNA. *n* = 3 biological replicates for each condition. (*Middle*) 24 h after 2.56 μM CREB inhibitor 666-15 treatment relative to cells treated with DMSO. *n* = 6 biological replicates for each condition. (*Right*) 24 h after transfection of a pCMV-CREB1 overexpression vector, relative to transfection with a dsRed-EGFP plasmid. *n* = 6 biological replicates. Same box plot definitions as in Fig. 1. Source data are provided as a Source Data file, as are exact *p*-values. * denotes *p* < 0.05, ** denotes *p* < 0.01, and *** denotes *p* < 0.001 for two-sided *t*-test comparison between conditions. **c** Correlations of MAPK/ERK/CREB1 stimulated transcript expression levels (nm. cts. is normalized read counts) with *CTR1* levels from healthy

human tissue (GTEx) transcriptomic data. Correlation line shadow corresponds to 95% confidence interval. Not significant correlations (*p* > 0.05) covered in gray boxes. Fit statistics provided in Supplementary Table 3. *n* = 578, 861, 2642, 226, 89 GTEx RNA-seq samples for lung, heart, brain, liver, and kidney tissues, respectively. **d** Boxplots generated from TCGA gene expression data of select MAPK/ERK/CREB1 stimulated transcripts, comparing MAPK/ERK activating KRAS^G12X mutant relative to non-variant lung adenocarcinoma transcript levels. Box plot values defined as follows: box boundaries (median of first quartile to median of third quartile) and central horizontal line (median of all expression values), numbers below/above individual box plots denote mean expression value of that dataset (shown on the plot as an "x"), whisker boundaries define 1.5 times the interquartile range; outlier data points omitted for visual clarity. *n* = 362 and 131 samples for KRAS non-variant and KRAS^G12X samples, respectively. * denotes *p* < 0.05, ** denotes *p* < 0.01, and *** denotes *p* < 0.001 for two-sided *t*-test comparison between conditions.

(Supplementary Fig. 11). Of further particular note, the copper treated dataset revealed enrichment of the response to cAMP ontology, consistent with a role of copper in activating CREB.

While MAPK3 (a.k.a. ERK1) target genes were significantly depleted upon TM treatment, CREB target genes were significantly enriched in the copper supplemented dataset (Fig. 3e,f). Indeed, copper generally stimulated activation of known CREB1 target genes, whereas TM generally repressed them (Fig. 3g). Finally, *CTR1* levels were significantly increased following TM treatment, further confirming that

decreasing bioavailable extracellular copper drives *CTR1* expression (Fig. 3a and Supplementary Fig. 12).

**The mechanism underlying copper stimulated EGFR activation**
To probe the molecular and biochemical mechanism underlying copper stimulated EGFR activation, we next applied anti-EGFR phosphorylation arrays to briefly serum-starved (~3 h) A549 cell lysates generated from cells supplemented with 20 μM CuCl₂ for 15 min, relative to unsupplemented cell lysates. An unambiguous,

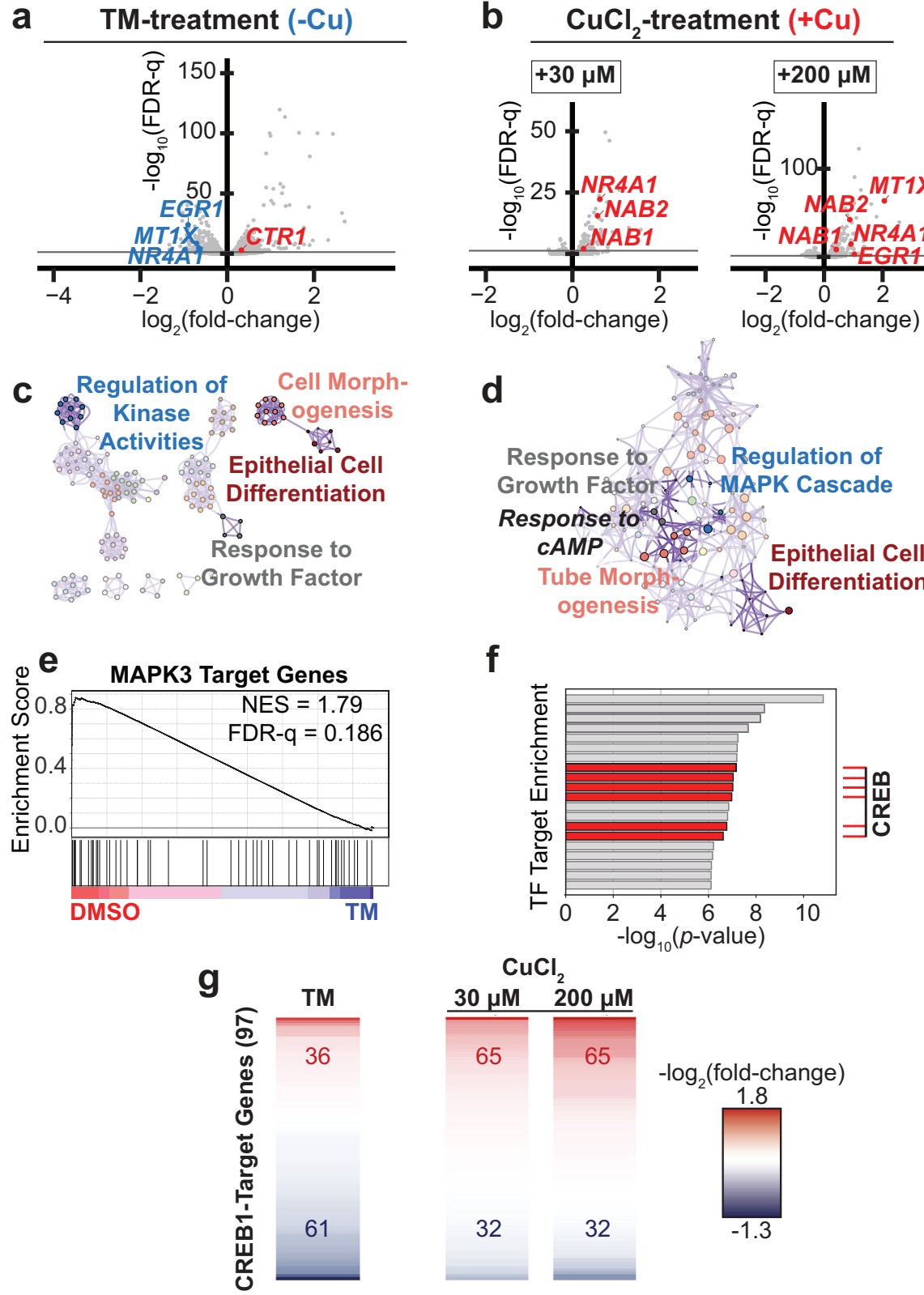

reproducible -50% decrease in pan-EGFR (total EGFR) levels in response to copper was measured (Supplementary Fig. 13). Rapid (sub-hour timescale) degradation of EGFR is an indicator of a specific interaction with EGFR, generally a conformational change driven by agonist binding; in contrast to the slow degradation induced by stress conditions[42]. Note that *EGR1* expression was activated by copper supplementation in cells pretreated with metalloproteinase inhibitor

to a similar magnitude as non-pretreated cells (Supplementary Fig. 14), excluding the possibility copper supplementation leads to activation of metalloproteinases which then cleave pro-growth factors and activate EGFR.

Given the rapid EGFR degradation upon copper supplementation, it was important to evaluate the possibility that extracellular $Cu^{2+}$ may directly bind EGFR to activate signaling. To probe $Cu^{2+}$ binding by

**Fig. 3 | Transcriptomic changes induced by altered copper levels and/or MAPK/ERK/CREB signaling.** RNA-seq volcano plots depicting differential expression for **a** MDA-MB-468 cells treated with TM for 24 h (relative to DMSO treated) and **b** A549 cells treated with CuCl₂ (30 μM or 200 μM) for 2 h (relative to unsupplemented). For visual clarity, outlier datapoints omitted from volcano plots; full plots shown in Supplementary Fig. 12. As expected, *MT1X* expression levels (which are correlated to intracellular copper levels following copper supplementation[55,56]) were decreased following TM treatment and increased following copper treatment. FDR-q is the false discovery rate adjusted *p*-value (corrected for multiple observations from the two-sided *p*-value calculated via Wald test by DESeq2). Gray lines denote FDR-q = 0.05. *n* = 2 (MDA-MB-468) or 3 (A549) biological replicates for each condition. Pathway and process enrichment for significantly differentially expressed transcripts following **c** TM or **d** 30 μM CuCl₂ treatment. All enriched ontologies provided in Supplementary Data 4–5. **e** Transcription factor target gene set enrichment analysis (GSEA) depicts the activation of MAPK3 (a.k.a. ERK1) target genes in DMSO-treated relative to TM treated cells; NES is normalized enrichment score. **f** TF (transcription factor) target enrichment generated from significantly differentially expressed transcripts post-2 h, 30 μM CuCl₂ treatment. Multiple CREB hits correspond to different CREB target gene lists. Fully labeled TF target enrichment presented in Supplementary Fig. 22. **g** Heatmaps depicting differential expression of CREB1 target genes (97 assessed in total) following TM or CuCl₂ treatment. Numbers in red and blue denote upregulated and downregulated CREB1 target genes, respectively. Gene list with corresponding expression values presented in Supplementary Fig. 23. The figure is intended to provide a general assessment of the treatment effects on CREB1 target gene expression−discrete lines (corresponding to individual transcripts) should not be compared across conditions and do not necessarily correspond to the same transcript.

EGFR, we expressed and purified soluble EGFR extracellular domain (sEGFR; residues 1-642). While we confirmed sEGFR bound $Cu^{2+}$ with a physiologically relevant $K_D$ (200−700 nM; Supplementary Fig. 15), and formed a well structured $Cu^{2+}$ binding site as assessed by electron paramagnetic resonance spectroscopy (Supplementary Fig. 16); no $Cu^{2+}$ driven dimerization (as required for ligand dependent EGFR signal transduction[43]) of sEGFR was detected by mass photometry (Supplementary Fig. 17). This contrasts with the recent speculation that $Cu^{2+}$ may drive dimerization of EGFR to activate signal transduction[44]. Having excluded copper stimulated EGFR dimerization as a potential activation mechanism of EGFR signal transduction, we explored a different possibility: that fluctuations in extracellular copper levels could lead to $Cu^{1+}$ import and inactivation of EGFR phosphatases, thereby effectively inducing EGFR activation.

The highly homologous (99% catalytic site similarity[45]) proteins PTPN1 and PTPN2 (a.k.a. PTP1B and TCPTP, respectively) are EGFR pY-1068 (and probably pY-1086[46]) phosphatases[47–49]. Previous studies assessing ex vivo[50,51] and in vitro (cellular)[44,52] environmental exposure to aberrant levels of copper have suggested metallo-inactivation of PTPN1, via an unknown molecular mechanism, and prior biochemical characterization of PTPN1 revealed strong inhibition of its phosphatase activity by a variety of metals (although $Cu^{1+}$ was not assessed)[53]. Relatedly, it was previously shown that PTPN1 overexpression was sufficient to partially block EGFR phosphorylation induced by cell membrane permeable copper complexes in glial cells[28]. These observations prompted us to consider PTPN1 and PTPN2 as alternative candidates for the copper sensor(s) responsible for relaying changes in intracellular copper levels into EGFR signaling responses. We subsequently screened siRNA knockdown of several PTPs−including established EGFR phosphatases PTPN1, PTPN2, PTPRJ and PTPRG[47]−and assessed copper stimulated EGFR activation by measuring *EGR1* expression level changes. *PTPN2* knockdown significantly diminished copper-stimulated *EGR1* activation in A549 cells; the only *PTP* knockdown to do so (Fig. 4a). This result supports a mechanism of copper driven activation of EGFR signaling via PTPN2 inactivation in A549 cells. We suspect the weakly decreased *EGR1* levels across most *PTP* knockdowns relative to negative control siRNA transfection are likely due to off-target effect(s) of the negative control siRNA, leading to slightly elevated *EGR1* levels. Indeed, the *DMT1* siRNA knockdown experiment also detected similarly decreased *EGR1* expression levels relative to negative control siRNA transfection (Supplementary Fig. 7).

For PTPN2 to function as a copper sensor, it should be more sensitive to fluctuations in intracellular copper levels than established copper-stress response biomarkers. Elevated expression of the metallothionein *MT1X* is an established biomarker of intracellular copper stress driven by copper supplementation[54–56]. We consequently evaluated changes in A549 *MT1X* expression via qPCR for the same CuCl₂ supplementation concentrations and incubation times as assessed previously. Only copper levels ≥100 μM drove elevated *MT1X* expression (Supplementary Fig. 18), far greater than the copper supplementation levels at which we previously detected significant EGFR activation and *EGR1* expression (20 μM and 10 μM CuCl₂, respectively) through PTPN2.

PTPN1 and PTPN2 dephosphorylate signal transduction proteins to attenuate signaling via a highly acidic active site cysteine. While $Cu^{2+}$ and other divalent metals are known to inactivate PTPN1, to the best of our knowledge, no biochemical assessment of $Cu^{1+}$ binding nor inactivation of PTPN1 has been performed[57–59]. As the PTPN1 and PTPN2 catalytic domains are highly homologous, and considerably more metallo-inactivation experiments have been performed on PTPN1, we recombinantly expressed and purified the PTPN1 catalytic domain (ΔPTPN1; residues 1-301), and subsequently evaluated $Cu^{1+}$ binding. We found that addition of excess $Cu^{1+}$ to ΔPTPN1 resulted in the protein binding of 5.4 ± 0.47 (*n* = 3) equivalents of copper, as measured by ICP-MS, and ΔPTPN1 activity was inhibited following incubation with $Cu^{1+}$ (as assessed by the *p*-nitrophenyl phosphate (*p*NPP) assay, Fig. 4b). The active site cysteine mutant C215S ΔPTPN1 bound 4.6 ± 0.26 (*n* = 3) $Cu^{1+}$ equivalents, indicating that the active site cysteine residue binds $Cu^{1+}$, and that the mechanism of the $Cu^{1+}$ driven inhibition involves $Cu^{1+}$ binding to the active site cysteine. By contrast, mutation of C121S (a cysteine residue not directly involved in enzyme catalysis[60]) induced no change in $Cu^{1+}$ binding stoichiometry (5.6 ± 0.14, *n* = 3). Experiments probing $Cu^{1+}$ binding by ΔPTPN2 and the corresponding ΔPTPN2 C216S active site mutant (in the presence or absence of 10 protein equivalents of the $Cu^{1+}$ chelator glutathione) yielded similar results. In both cases, the ΔPTPN2 $Cu^{1+}$ binding stoichiometry was greater than that of C216S ΔPTPN2 (no glutathione: 1.9 ± 0.16 and 1.8 ± 0.13, respectively; with glutathione: 3.2 ± 0.36 and 3.0 ± 0.24, respectively; *n* = 3 for all). The increased $Cu^{1+}$ binding stoichiometry in the presence of $Cu^{1+}$ chelator may be a consequence of glutathione stabilizing $Cu^{1+}$ binding to the protein. We therefore confirmed that copper ions can biochemically modulate PTPN1/2 activity via binding to the catalytic active site cysteine.

We then sought to evaluate specificity of copper driven PTP inhibition, first relative to the chemically similar, bioavailable metal zinc. We detected no significant increase in CREB nor EGFR phosphorylation following addition of the same concentration of ZnCl₂ as added for CuCl₂ (20 μM) (Fig. 4c, Supplementary Fig. 19). Moreover, we found ZnCl₂ supplementation drove weaker activation of *EGR1* expression (and therefore, weaker EGFR activation) than CuCl₂ in A549 cells (Fig. 4d). Indeed, previous studies of multiple cell lines found that while cell membrane permeable $Cu^{2+}$ complexes induced strong EGFR activation, the $Zn^{2+}$ complexes (or ZnCl₂ supplementation) induced little to no effect[28,61]. Thus, PTP inactivation is more sensitive to copper level fluctuations than zinc, particularly in the physiological range of concentrations.

Finally, to confirm that cytosolic copper levels modulate the PTPN2-EGFR pathway, as well as to provide evidence that copper inhibition of PTPN2 occurs under normal cellular growth conditions, we supplemented A549 cells with chemical probes to directly

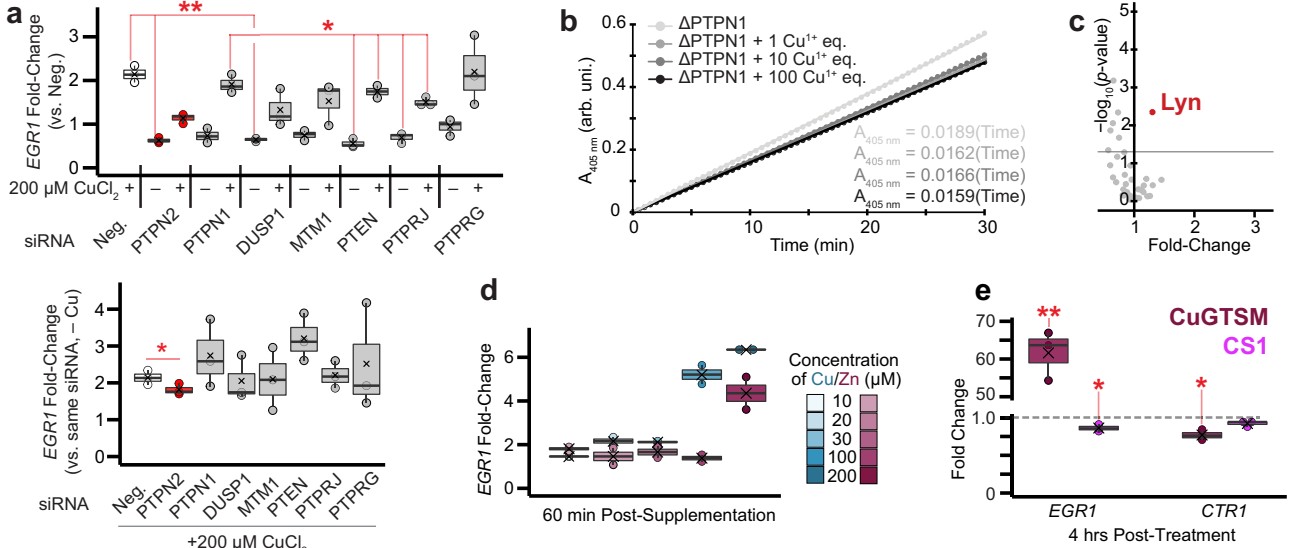

**Fig. 4 | Inhibition of PTPN2 is associated with copper stimulated activation of EGFR. a** Evaluation of the effects on *EGR1* expression levels of 200 μM CuCl₂ (60 min treatment) following siRNA knockdown of various PTPs in A549 cells. Depicted are two interpretations of the results. (*top*) All fold-changes are reported relative to the negative control siRNA transfect (Neg.) without CuCl₂ supplementation. (*bottom*) Each result is reported as the fold-change between the denoted siRNA knockdown after 60 min 200 μM CuCl₂ supplementation and the siRNA knockdown by itself. (*n* = 3 replicates per condition). Same box plot definitions as in Fig. 1. * denotes *p* < 0.05, ** denotes *p* < 0.01, and *** denotes *p* < 0.001 for two-sided *t*-test comparison between conditions. **b** *p*NPP (*p*-nitrophenyl phosphate) assay of ΔPTPN1 PTP activity following addition of various amounts of Cu¹⁺ (*n* = 12 replicates per condition). **c** Proteome profiler antibody array volcano plot depicting

statistically significant increases in phosphorylation levels following HEK 293 T supplementation with 20 μM ZnCl₂. (*n* = 2 replicates for each condition). **d** Changes in A549 *EGR1* expression levels 60 min after ZnCl₂ supplementation. The fold-changes are overlaid on the fold-changes induced by CuCl₂ supplementation for reference. (*n* = 2 replicates per condition). Same box plot definitions as in Fig. 1. **e** RNA-qPCR fold-changes of select transcripts 4 h post-supplementation of 500 nM CuGTSM or 5 μM CS1 relative to DMSO treated A549 cells. (*n* = 3 replicates per condition). Data points correspond to biological replicates, *p*-values calculated as paired two-tailed *t*-test. Same box plot definitions as in Fig. 1. Figure 4a, b, d, and e source data are provided as a Source Data file, as are exact *p*-values for Fig. 4a, e. * denotes *p* < 0.05, ** denotes *p* < 0.01, and *** denotes *p* < 0.001 for two-sided *t*-test comparison between conditions.

elevate or depress cytosolic copper levels (bypassing CTR1/DMT1 and intracellular Cu-chaperones). Specifically, we added the copper ionophore Cu-GTSM (which transports Cu²⁺ across the plasma membrane and releases Cu¹⁺ in the cytosol) or the cytosolic copper sensor CS1 (to effectively decrease labile intracellular Cu¹⁺ levels via binding competition) and assessed changes in EGFR signaling activation via RNA-qPCR (relative to cellular treatment with DMSO control). Four hours after Cu-GTSM supplementation, *EGR1* expression was strongly elevated while *CTR1* levels were decreased even more strongly than CuCl₂ treatment alone (-25% decrease relative to control, Fig. 4e). Four hours after CS1 treatment, by contrast, *EGR1* levels significantly decreased, consistent with labile copper driving PTPN2 inhibition under normal cellular growth conditions. For completeness, *CTR1* expression levels were not significantly affected by CS1 treatment.

We then evaluated specificity of copper driven PTP inhibition relative to general, non-specific PTP inhibition. We compared the CuCl₂ treated A549 cell proteome profiler antibody array results to signal transduction activation following addition of 20 μM pervanadate solution, which serves as a general PTP inhibitor (via oxidation of PTP active site cysteines). While copper supplementation drove significant repression of AKT and p70S6K phosphorylation, presumably indicating repression of mTOR signaling, pervanadate (perhaps unsurprisingly) drove activation of virtually all signaling proteins evaluated (Supplementary Fig. 20). Thus, the copper signaling responses elicited by PTPN2 inactivation in A549 cells do not reflect a broad PTP inhibition mechanism, but rather drive more specific signaling activation.

## Discussion

Toxicological studies assessing ex vivo[50,51] and in vitro (cellular)[44,52] environmental exposure to aberrant levels of copper proposed metallo-inactivation of the ubiquitous, highly expressed PTPN1 via an

unknown molecular mechanism. The previous literature offers conflicting reports on whether or not low level copper concentration fluctuations—such as occurring under normal, physiological conditions—affect this pathway[28,44,61]. Relatedly, our findings represent two main discoveries.

First, we have identified PTPN2 as a mammalian copper receptor capable of relaying changes in physiological copper levels into signal transduction responses. Regarding the apparent copper inactivation specificity of PTPN2 over PTPN1, a more comprehensive analysis of *PTP* siRNA knockdowns in MCF7 breast cancer cells also found *PTPN2* knockdown significantly diminished EGFR activation by EGF, while *PTPN1* knockdown had no effect[48]. Cell imaging experiments found PTPN1 localized almost exclusively to the perinuclear region, in contrast to PTPN2, which was more diffuse throughout the cell[48]. Thus, we propose PTPN2 is a more sensitive reporter of changes in imported copper levels due to its more proximal localization near the cell membrane (and membrane localized EGFR). We therefore assign PTPN2 as the physiological sensor of copper but note that due to the highly homologous nature of PTPN1 and PTPN2, we do not exclude the possibility that PTPN1 can also function in such a role in different cellular contexts. Indeed, following knockdown of *PTPN2*, copper treatment still stimulated activation of *EGR1* expression, albeit to a lesser extent than without knockdown, reminiscent to similar results reported following *PTPN2* knockdown and EGF activation of EGFR[48]. As receptor tyrosine kinases like EGFR are regulated by a network of PTPs[48,62], this result implies that other EGFR-regulating PTPs may also be susceptible to copper inhibition.

Second, we have established a link between CREB activation and weak *CTR1* transcriptional suppression (Fig. 5). Given the super-physiological copper supplementation levels required to drive significant *CTR1* transcriptional repression, it seems implausible that this mechanism exists purely to regulate subtle fluctuations in

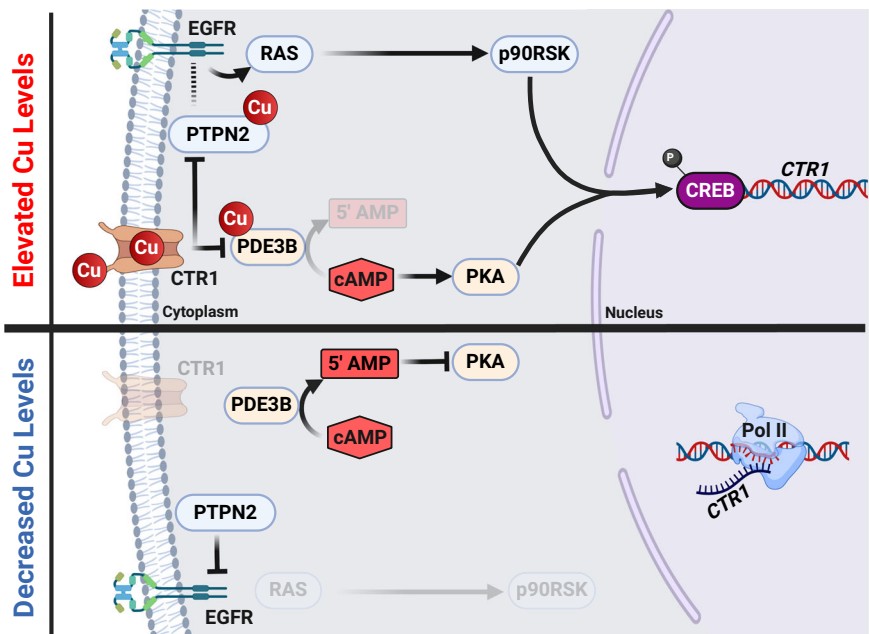

**Fig. 5 | Schematic representation of copper signaling and *CTR1* repression.** (top) Proposed mechanism of copper-stimulated EGFR activation, with downstream CREB TF activity and transcriptional repression of *CTR1*. (bottom) Decreased Cu levels result in downregulation of the pathway shown on the top. Figure 5 created with BioRender.com released under a Creative Commons Attribution-NonCommercial-NoDerivs 4.0 International license.

physiological copper levels, and consequently, our results do not contradict the current mechanistic model that the dominant mechanism of copper stimulated CTR1 expression regulation occurs at the post-translational level[15]. However, processes such as neuronal differentiation (as modeled by PC12 cells treated with nerve growth factor, NGF) involve strong, sustained CREB activation[63], substantial intracellular copper accumulation[6,20], and strong *CTR1* transcriptional repression (via an unknown regulatory mechanism)[6]. Thus, we propose CREB activation may be the underlying driver of this *CTR1* repression. Intriguingly, prior studies of PC12 cells have elucidated additional connections between NGF, copper, and CREB: NGF (or at least a peptide corresponding to the NGF N-terminal sequence) can serve as a copper ionophore, and copper and NGF co-supplementation drives stronger activation of CREB than either component individually[64,65].

The Zn finger TF SP1 regulates *CTR1* transcription[66]; copper stimulated changes in *CTR1* transcript levels have been previously reported and attributed to copper responsive modulation of SP1 TF activity[17,67–69]. In the reported mechanism, intracellular copper stress drives copper binding and displacement of $Zn^{2+}$ ions from the SP1 zinc fingers[70,71], inhibiting TF activity and thereby lowering *CTR1* (and other SP1 target gene, e.g., *SP1*, *FOXM1*)[68,72] expression as SP1 is generally a transcriptional activator[73]. As copper chelation elevated *CTR1* levels in these studies, the model implies the presence of "copper substituted" SP1 under normal cellular growth conditions.

However, this model may need to be reconciled with some experimental observations. Namely, copper deficiency in mouse models does not alter *Ctr1* expression levels across multiple organs[16]. Additionally, with over 12,000 SP1 binding sites in the genome[73], copper inactivation of SP1 would drive broad repression of many genes unrelated to copper homeostasis (including various housekeeping genes), implying a lack of specificity in the response. Relatedly, in NSCLC cell lines (including A549), SP1 TF activity drives constitutive *VEGF* expression[74]; thus, by the SP1 copper substitution model, copper stress should suppress *VEGF* expression. By contrast, copper is an established pro-angiogenic factor known to stimulate expression of

*VEGF*[75], which we also detected via RNA-seq 4 h after super-physiological copper level supplementation (*VEGFA*, Supplementary Data 3). Additionally, we found no statistically significant change in *FOXM1* expression levels following copper supplementation at any concentration, while *SP1* expression was significantly increased 4 h after super-physiological copper level supplementation. Future work will be required to understand how the model we have established in this report fits with the SP1/*CTR1* regulatory model. It may be that copper inhibition of PTPN2 and/or SP1 is cellular context dependent: reflective of the relative levels of various $Cu^{1+}$ binding proteins, the level of copper stress, and/or the means through which copper stress is elicited.

Interestingly, both the pathway reported here and copper driven inhibition of PDE3B (which leads to elevated cAMP levels, as known to drive CREB activation) converge on cAMP and CREB regulation to connect copper availability with cell growth, proliferation, and differentiation. Finally, EGFR, CREB, and serum/tissue copper depletion are all targets in anti-cancer treatments. Our discovery of a signaling axis involved in *CTR1* repression (a known cause of platinum anticancer drug resistance) represents discovery of a potential targetable axis for synergistic drug efficacy.

## Methods
Unless otherwise noted, all materials were obtained from Thermo-Fisher. A549 cells were purchased from ATCC (catalog number: A549 (ATCC® CCL-185™)), while HEK 293 T were acquired from a lab member who acquired them from ATCC. Cetuximab antibodies were a kind gift from Dr. Bruce Marc Bissonnette. CREB Inhibitor 666-15 was purchased from Sigma-Aldrich and dissolved in DMSO to establish a 17.14 mM stock. All volcano plots were generated in RStudio. The term "biological replicate" indicates discrete, separate samples; the same sample was never measured repeatedly. Silencer® Select siRNAs were purchased from ThermoFisher with the following product lot and IDs: PTPN1 (AS02KZQB, s11507), DUSP1 (AS02KZQF, s4363), PTPRG (AS02KZQD, s11549), PTPN2 (AS02KZQC, s11509), MTM1 (AS02KZQH, s9041), PTPRJ (AS02KZQE, s11555), PTEN (AS02KZQG, s536620),

CREB1 (AS02JUZ3, s3490), CTR1 (AS02MHH1, s3377), DMT1 (AS02MHH0, s9708), Negative Control #1 siRNA (AS02J03Q, no ID provided).

## Cell culture

A549 and HEK 293 T cells were cultured at 37 °C, 5% CO₂, 90% humidity in CellXpert C170i Cell Culture Incubators. Culture media was Dulbecco's Modified Eagle Medium (DMEM) supplemented with 10% Fetal Bovine Serum (FBS) and 1% Penicillin-Streptomycin (10,000 U/mL) (this combined solution is referred to as supplemented-DMEM, sDMEM). Sub-culturing was performed by aspirating the media, washing with an equivalent volume of Dulbecco's phosphate-buffered saline (DPBS), aspirating the DPBS, washing with one-fifth the original media volume 0.25% Trypsin-EDTA, aspirating, incubating for 1 min in the cell culture incubator, and then resuspending in sDMEM. Cells were seeded at a 1:4 ratio of cell resuspension solution to fresh sDMEM, and were maintained under 95% confluency. Total sDMEM volume used for 35 mm (6 well plate), 100 mm, and 150 mm Corning cell culture plates was 2 mL (per well), 10 mL, and 20 mL, respectively. Cells were verified mycoplasma negative using the MycoProbe Mycoplasma Detection Kit (R&D Systems). For serum-starvation experiments, cell culture sDMEM was replaced with DMEM for times indicated in figures, and cells were subsequently either treated with pre-warmed (to 37 °C) DMEM or DMEM supplemented with CuCl₂ at the stated concentrations. For gefitinib/cetuximab treatments, cells were pre-treated with either 2 μM gefitinib in sDMEM for 3 h or 80 μg/mL Cetuximab in sDMEM for 4 h in the cell culture incubator. For subsequent Cu-treatment of EGFR-inhibitor pretreated cells: Cu-supplemented, EGFR-inhibitor supplemented (same concentrations) sDMEM was pre-warmed (to 37 °C) and subsequently used to replace the cell supernatant media. For Cu-GTSM (Cayman Chemical Company) and CS1 (MedChemExpress) treatments, the solid drugs were dissolved in DMSO to a final concentration of 1 mM. 10 μL, 100 μL, and 100 μL of Cu-GTSM, CS1, and DMSO solutions, respectively, were added to 20 mL of prewarmed sDMEM. These solutions were then used to replace the cell supernatant media, and the cells were incubated in the cell incubator for 4 h before RNA extraction.

MDA-MB-468 cells were cultured by Dr. Olufunmilayo Olopade's group in RPMI-1640 supplemented with 10% FBS and antibiotic-antimycotic solution (Gibco). Cells were handled and treated (with TM- and/or DMSO) as described elsewhere[76].

## Pervanadate generation

To generate the pervanadate solution for PTP inhibition, we first generated a fresh stock of 100 mM sodium orthovanadate in water. 20 μL of this solution was mixed with 23 μL 30% H₂O₂ and 157 μL of DMEM, which was then incubated at room temperature (in the dark) for 30–45 min to prepare a 10 mM pervanadate stock. Pervanadate solutions were generated immediately prior to cell treatment.

## Antibody array and associated imaging

Cells were seeded in 150 mm plates and treated at ~80% confluency. Following serum-starvation, supernatant media was removed and replaced with either DMEM or metal-supplemented DMEM. The plates were subsequently put back into the cell culture incubator for the times noted in the text. Cells were harvested, lysed, and lysates assessed with the Proteome Profiler Human Phospho-Kinase Array Kit (R&D Systems) and C-Series Human EGFR Phosphorylation Antibody Array 1 Kit (Ray-Biotech) according to manufacturer protocols. Imaging was conducted with the FluorChem R System; experimental and control membranes were imaged for identical exposure durations to facilitate normalization of chemiluminescence intensities across membranes. Integrated pixel densities were measured in ImageJ (1.53k); pixel densities were subsequently background corrected and normalized to reference control spot-antibody pixel density intensities.

## ChIP-seq data analysis

ChIP-seq BigWig files were accessed from the ChIP-Atlas database. Wiggletools v1.2.11 was used to calculate the mean of BigWig files as indicated in the text, which was output as a Wig file. Wig files were subsequently converted to BigWig files using the ucsc-wigtobigwig package (v377). BigWig files were visualized using IGV.

## KAS-seq and data analysis

The KAS-seq labeling and isolation procedure was followed as previously described[77], with the following changes. Cell culture media was removed and subsequently replaced with CuCl₂-supplemented (or CuCl₂-unsupplemented control) sDMEM for times indicated in the volcano plots. Following incubation, 500 mM N₃-kethoxal (in DMSO) stock was added directly to the cell media to a final concentration of 5 mM. After addition, the solution was vigorously shaken to promote solvation of the moderately-soluble N₃-kethoxal. Dual index library construction used an Accel-NGS Methyl-Seq DNA library kit; libraries were sequenced at the Genomics Facility (University of Chicago) via single-end Illumina NovaSeq 6000 (SP flowcell, 100 bp cassette) sequencing. Reads from two sequencing runs were catenated, and then processed following the KAS-pipe data processing pipeline[77] (mapped to human genome build hg19) on a Lenovo Thinkstation P920 workstation.

## RNA extraction and RNA-qPCR

Total RNA was isolated using either TRIzol according to provided instructions, or a combination of TRIzol and the RNA Clean & Concentrator-5 (RCC-5) kit (Zymo Research), according to the Zymo instructions for combination Trizol/Zymo-Spin-Column RNA isolation. Samples prepared using the RCC-5 kit were also treated with in-column DNase I following Zymo instructions. cDNA synthesis was performed via either Maxima First Strand cDNA Synthesis Kit for RT-qPCR (with dsDNAse added), or High Capacity cDNA Reverse Transcription Kit (Applied Biosciences). RNA levels were quantified by determining cycle threshold (Ct) values of corresponding cDNAs with the Quant-Studio 6 Pro system (Applied Biosciences, A43180) and FastStart Essential DNA Green Master (Roche) at a final concentration of 1X, with ~500 nM forward and reverse primers, in a final reaction volume of 20 μL per well in MicroAmp Optical 96-Well Reaction Plates (primer sequences provided in Supplementary Table 4). Samples were amplified with the following run method: 10 min hold at 95 °C, 40x cycles of 20 s at 95 °C, 20 s at 60 °C, and 20 s at 72 °C (during which fluorescence measurements occurred), before a final 15 s at 95 °C and 1 min at 60 °C. The cycle was completed with a melting curve while measuring fluorescence. All biological replicates comprised at least 2 technical replicates. All transcripts were processed using the $2^{-\Delta\Delta Ct}$ relative expression normalization method[78], using *GAPDH* as an internal reference standard.

Copper treatment RNA-qPCR experiments compared metal-treated to control sDMEM-treated values. siRNA treatment experiments compared cDNA generated following experimental siRNA treatment to cDNA generated from Silencer® Select Negative Control No. 1 siRNA-transfected cells. Gefitinib pretreatment experiments compared gefitinib-pretreated, Cu-treated to gefitinib-pretreated cell cDNA. Cu-GTSM and CS1 experiments compared drug treated cell cDNA to DMSO-treated cell cDNA. Two-tailed, paired T-test *p*-values were calculated in RStudio, using the following packages: dplyr v1.0.7, tidyverse v1.3.1, readxl v1.3.1, ggprism v1.0.3. Example code is provided in the Supplementary Materials.

## *CREB1* and *PTP* Knockdown

Knockdown of CREB1 and PTPs was performed via transfection of Silencer Select siRNAs using Lipofectamine™ RNAiMAX Transfection Reagent. Specifically, cells were seeded in sDMEM lacking antibiotics ~24 h prior to transfection to generate 6 well plates in which individual

wells were ~40–50% confluent at time of treatment. Per cell treatment: 9 μL of Lipofectamine was diluted into 150 μL of Opti-MEM at room temperature, and separately, 3 μL of 10 μM siRNA was diluted into 150 μL of Opti-MEM. The diluted siRNA was then mixed with the diluted Lipofectamine in a 1:1 ratio, and incubated at room temperature for 5 min. 250 μL of the Lipofectamine/siRNA mixture was added to the 2 mL of cell media in the well, and then quickly mixed back and forth to promote homogenization. Cells were then put back into the cell incubator for 24 h prior to treatment and/or RNA extraction and RNA-qPCR measurement.

## Constitutive CREB expression vector transfection

The CREB Dominant-Negative Vector Set was purchased from Takara Bio, and the vectors subsequently transformed into DH5α Mix and Go! competent *E. coli* (Zymo Research) and plated onto LB-agar plates with 50 μg/mL kanamycin and colonies grown overnight at 37 °C. Vector transformed colonies were picked and grown in a 10 mL LB culture with 50 μg/mL kanamycin (37 °C with vigorous shaking). This culture was used to inoculate overnight 1 L cultures of LB with 50 μg/mL kanamycin (grown at 37 °C with vigorous shaking). The plasmid was subsequently purified using a QIAGEN plasmid maxi kit following manufacturer protocol.

A549 cells in antibiotic-free sDMEM were transfected with 2.5 μg of purified pCMV-CREB Vector (corresponding to a plasmid for constitutive CREB expression) using a Lipofectamine 3000 reagent kit, following the manufacturer protocol. As a negative control, cells were transfected (following the same procedure) with the DsRed-EGFP in a pcDNA3.1 HisC vector[79].

## DMT1 constitutive expression vector transfection and siRNA knockdown

The DMT1/SLC11A2 ORF was cloned into a pcDNA3.1 + /C-(K)DYK vector by Genscript. The vector was subsequently transformed into One Shot MAX Efficiency DH5-T1R Competent *E. coli* cells via heat shock and were plated onto Amp100 plates (100 μg/mL ampicillin in LB-agar) for antibiotic selection. Vector-transformed *E. coli* colonies were subsequently picked and grown at 37 °C in 10 mL starter LB-cultures with 100 μg/mL ampicillin. This cell suspension was then used to inoculate 600 mL of LB media + 100 μg/mL ampicillin; this culture was grown with vigorous shaking at 37 °C overnight. The plasmid was subsequently purified using a QIAGEN plasmid maxi kit following manufacturer protocol.

A549 cells in antibiotic-free sDMEM (in 6 well plates with 2 mL of media) were transfected with 2.5 μg of the purified DMT1 vector using a Lipofectamine 3000 reagent kit, following the manufacturer protocol. As a negative control, cells were transfected (following the same procedure) with the DsRed-EGFP in a pcDNA3.1 HisC vector[79]. In parallel, DMT1 Silencer Select siRNA transfections were performed following the same procedure as described for the *CREB1/PTP* knockdown. 48 h after vector or siRNA transfection, cells were treated with 200 μM $CuCl_2$. Cells were incubated in the cell incubator for 1 h (37 °C, 5% CO2, 90%) before RNA extraction and cDNA generation.

## MDA-MB-468 RNA-seq and data analysis

Total RNA was extracted from MDA-MB-468 cells via Trizol extraction; the RiboMinus Eukaryote Kit was then used according to manufacturer protocol to remove ribosomal RNA. RNA were then used for library construction using a SMARTer Stranded RNA-seq kit (Takara, 634839) according to the manufacturer protocol. Paired-end high-throughput sequencing was subsequently performed at the University of Chicago Genomics Facility on an Illumina HiSeq 4000 sequencer.

Reads were trimmed using the KAS-pipe paired-end trim_adapter.sh script. Trimmed reads were aligned to STAR reference genomes (using STAR version 2.7.9a) generated from the human genome build hg38. Aligned reads were then processed to remove (i) rows

“N_ambig”, “N_multima”, and “N_noFeat” and (ii) transcripts with 2 or fewer average read counts. Unstranded column reads were then processed for differential expression analysis via DESeq2 v1.36.0. Pathway/process enrichment analysis of significantly differentially-expressed transcripts was performed via Metascape[80], while GSEA was performed using GSEA v4.2.3.

## A549 RNA-seq and data analysis

Total RNA was isolated using a combination of TRIzol and the RCC-5 kit (Zymo Research), according to Zymo's instructions for combination Trizol/Zymo-Spin-Column RNA isolation. Samples prepared using the RCC-5 kit were also treated with in-column DNase I according to provided Zymo instructions. Total isolated RNA was subjected to two rounds of poly(A)-enrichment using the Dynabeads mRNA DIRECT Kit according to the manufacturer's instructions twice on each sample.

Library preparation was performed with 5 μL of twice poly(A)-enriched sample using the NEBNext Ultra II Directional RNA Library Prep Kit for Illumina (New England BioLabs) according to manufacturer's instructions for "Protocol for use with Purified mRNA or rRNA Depleted RNA". Paired-end high throughput sequencing was subsequently performed on an Illumina NextSeq 550. FastQ files from each sample across runs were subsequently combined. Adapter trimming and basic QC was carried out with Trim Galore. Alignment to the human hg38 reference genome was carried out with HISAT2 (v2.2.1). Duplicate read removal was carried out with Samtools (v1.13). Transcript identification and read counting was performed with feature-Counts according to gene transfer format files provided by release 107 of the Ensembl database.

Transcripts with fewer than 0.74 average reads per replicate across all samples were excluded from analysis. Differential expression analysis was performed by the DESeq2 R package DESeq2 v1.34.0. Pathway/process enrichment analysis of significantly differentially-expressed transcripts was performed via Metascape[80].

## TCGA and GTEx RNA-seq data analysis

Lung adenocarcinoma (LUAD) TCGA data was accessed via the Xenabrowser server[81]—transcript read counts were converted from "$\log_2$(norm count + 1)" to "norm count + 1" prior to statistical analysis. $p$-values were generated in Excel via the t-Test: Two-Sample Assuming Unequal Variances function.

GTEx Analysis V8 release gene read counts were downloaded from the GTEx Portal. Read counts were first separated by tissue into discrete datasets, which were subsequently processed and analyzed separately. Next, transcripts with 2 or fewer average read counts were removed. Next, differential expression analysis (comparing tissue samples against an arbitrarily chosen sample from the same tissue) via DESeq2 v1.36.0 was performed to generate normalized read counts. Transcript correlations and plots (as well as associated statistical analysis) were subsequently generated in RStudio from the DESeq2 normalized read counts using ggpubr v0.4.0. Example code is provided in the Supplementary Materials.

## Cloning and expression of sEGFR

Recombinant human EGFR extracellular domain (residues 1-642) was expressed in insect cells as secreted protein using an approach described previously[82]. Briefly, human EGFR extracellular domain (residues 1-642) followed by an HRV3C cleavage site and a C-terminal 6x-histidine tag was subcloned into a pVL1393 vector. To generate the baculovirus, the constructed plasmid was co-transfected with the linearized baculovirus DNA (Expression Systems, 91-002) using Cellfectin II (Thermo Fisher, 10362100). Baculovirus was subsequently amplified in Sf9 cells (Thermo Fisher, 12659017) cultured in Sf-900 III SFM medium (Gibco, 12658019), supplemented with 10% heat-inactivated fetal bovine serum (Cityva, SH30396), 2 mM L-Glutamine (Cityva, SH3003402), and 10 μg/mL gentamicin at 27 °C. Large-scale protein

expression was performed by infection of High-Five cells (Thermo Fisher, B85502) in Insect-XPRESS medium (Lonza, BELN12-730Q) medium at a cell density of $2.0 \times 10^6$ cells/ml for 64 to 70 h at 27 °C.

## Purification of sEGFR

The EGFR signal peptide (residues 1–24) was cleaved after secretion. The medium containing secreted EGFR (25–642) was collected and centrifuged at $900 \times g$ for 15 min at room temperature. The supernatant was transferred into a beaker and mixed with buffer and salt reagents; the final mixture contained 50 mM Tris, pH 8.0, 5 mM $CaCl_2$ and 1 mM $NiCl_2$. The mixture was stirred at 130 rpm for 30 min at room temperature. After a centrifugation ($8000 \times g$, 30 min), the cleared supernatant was incubated with Ni-NTA resin (Thermo Scientific, 88223) for 4 h at room temperature. The resin was collected using a glass Buchner funnel and rinsed with HBS buffer (10 mM HEPES, pH 7.2, 150 mM NaCl) containing 10 mM imidazole, before transferring to a gravity column (Bio-Rad, 7372512). The protein was eluted with HBS buffer containing 200 mM imidazole. The eluent was concentrated and further purified by size-exclusion chromatography (Superdex 200 10/300 GL; GE Healthcare) equilibrated in a buffer containing 20 mM HEPES, pH 7.5 and 100 mM NaCl. Cleavage of the His-tag by HRV3C protease (Takara, 7360) followed the manufacturer's protocol. HRV3C protease and cleaved His-tag were removed from sEGFR by a second size-exclusion chromatography (Superdex 200 10/300 GL) run. Purified sEGFR with or without His-tag were validated by western blot (Supplementary Fig. 21) and successful removal of His-tag was observed in HRV3C protease treated sEGFR using the anti-His-tag antibody (Genscript, A00186-100) at a final concentration of 0.5 µg/mL.

## Phen green competition experiments

For fluorescence measurements, 1 mg aliquots of Phen Green were dissolved in 472 µL of DMSO−from this stock, a 3 µM Phen Green stock in 20 mM HEPES, 100 mM NaCl, pH 7.5 was prepared. Invitrogen 96-well Fluorescence-based Assay Microplate wells were filled with 600 µL of either (i) 20 mM HEPES, 100 mM NaCl, pH 7.5 buffer control, (ii) 3.64 µM sEGFR, 1 µM Phen Green (dissolved in DMSO) in 20 mM HEPES, 100 mM NaCl, pH 7.5, and (iii) 3.64 µM sEGFR, 1 µM Phen Green in DMSO in 20 mM HEPES, 100 mM NaCl, pH 7.5 with a screen of $CuCl_2$ concentrations (denoted in the respective figures) ranging from 0 equivalents (relative to Phen Green) to 1 equivalent. Fluorescence measurements were performed on a Synergy HTX Multi-Mode Reader, using an excitation wavelength of 485 nm and an emission wavelength of 528 nm (with the optics in the top position). Following measurement, to all wells 3 µL of 5 µM $CuCl_2$ in 20 mM HEPES, 100 mM NaCl, pH 7.5 were added; samples were then incubated at room temperature for 15 min prior to subsequent fluorescence measurement. $\Delta_{Fluorescence}$ titration curves were then generated by averaging 5 measurements for each $CuCl_2$ titration concentration, and defining the 0 µM $CuCl_2$ added measurement as "0" and the lowest fluorescence intensity measurement as "1". Curve fitting was performed using the Matlab Curve Fitting App to the equation $\triangle_{Fluorescence} = \frac{B_{max}[CuCl_2]}{K_D{}^{app} + [CuCl_2]}$, where $[CuCl_2]$ corresponds to the added concentration of $CuCl_2$ in 20 mM HEPES, 100 mM NaCl, pH 7.5, $B_{max}$ corresponds to the number of Cu binding sites on the protein, and $K_D{}^{app}$ is the apparent Phen Green $K_D$ (corresponding to the true Phen Green KD when [sEGFR] = 0, assuming negligible $Cu^{2+}$ binding by the buffer) This value was used to determine the sEGFR $Cu^{2+}$ as described below.

We first assumed that there was no "free" solution $Cu^{2+}$: all $Cu^{2+}$ added to the solution was bound by sEGFR or Phen Green. Then, from the equation $\frac{[ML][P]}{[MP][L]} = \frac{K_{D(P)}}{K_{D(L)}}$; where [ML], [P], [MP], and [L] correspond to the Phen Green-bound $Cu^{2+}$, apo-sEGFR, sEGFR-bound $Cu^{2+}$, and apo-Phen Green concentrations, respectively, while $K_{D(P)}$ and $K_{D(L)}$ are the

$Cu^{2+}$ $K_D$ values for sEGFR and Phen Green, respectively; [L] = [ML] when [MP] = $K_D{}^{app}$ (this is just a substitution of [MP] for "free" solution $Cu^{2+}$, or [M] in this case, into the definition of $K_D$). Consequently, when [MP] = $K_D{}^{app}$, the equation may be simplified to $\frac{[P]}{[MP]} = \frac{K_{D(P)}}{K_{D(L)}}$ or $\frac{[P]}{K_D{}^{app}} = \frac{K_{D(P)}}{K_{D(L)}}$. From this equation, we calculated $K_{D(P)}$.

## ICP-MS

Metal-bound protein solutions (sEGFR, $\Delta$PTPN1 and associated mutant, $\Delta$PTPN2 and associated mutant) were precipitated to release bound metals via addition of 5% $HNO_3$. The protein precipitate was subsequently pelleted by centrifugation. Sample measurements were performed via an Agilent 7700x ICP-MS in He mode, and were subsequently analyzed using ICP-MS Mass Hunter version B01.03. Intensities were correlated to ppb concentrations using multi-element ICP-MS standards (Inorganic Ventures).

## EPR spectroscopy

For EPR measurements, ~200 µL of ~100–200 µM sEGFR was aliquoted into Wilmad quartz X-band EPR tubes and flash frozen in liquid nitrogen. X-band continuous wave (CW) EPR measurements were performed at both the University of Chicago using a Bruker Elexsys E500 spectrometer featuring an Oxford ESR 900 X-band cryostat with a Bruker Cold-Edge Stinger.

## Mass photometry

Mass photometry experiments were conducted at the University of Illinois at Chicago campus (Biophysics Core), using a Refeyn Two Mass Photometer (Refeyn Ltd, Oxford, UK). Data acquisition and processing were performed using Acquire MP software and DiscoverMP software, respectively. sEGFR in 20 mM HEPES, 100 mM NaCl, pH 7.5 (final concentration of 100 nM) were incubated overnight at 4 °C with the indicated concentrations of $CuCl_2$ (in 20 mM HEPES, 100 mM NaCl, pH 7.5) or 20 mM HEPES, 100 mM NaCl, pH 7.5 buffer as a control. 10 µL of protein sample was then mixed with 10 µL of 20 mM HEPES, 100 mM NaCl, pH 7.5 buffer on the sample well cassette before immediately beginning the measurement. Ratiometric contrast to mass conversion was performed by comparison to Thyroglobulin (TG) and beta-amylase (BAM) provided by the University of Illinois at Chicago Biophysics Core.

## Cloning and expression of $\Delta$PTPN1/2 constructs

The $\Delta$PTPN1, C215S $\Delta$PTPN1, $\Delta$PTPN2, and C216S $\Delta$PTPN2 constructs were cloned into a pET-28a(+)-TEV vector, featuring C-terminal TEV protease cleavage site followed by (His)$_6$-tag, by Genscript. The C121S $\Delta$PTPN1 mutant was generated from the $\Delta$PTPN1 vector using a Quik-Change II Site Directed Mutagenesis kit (Agilent), and the mutant sequence verified by whole plasmid sequencing (Azenta). Vectors were transformed into BL21 Star (DE3) *E. coli* via heat shock and were plated onto Kan50 plates (50 µg/mL kanamycin in LB-agar) for antibiotic selection. Vector-transformed *E. coli* colonies were subsequently picked and grown at 37 °C in 10 mL overnight starter LB-cultures with 50 µg/mL kanamycin. After overnight growth, 3 mL of cell suspension was used to inoculate 1 L of LB media + 50 µg/mL kanamycin. This culture was grown with vigorous shaking at 37 °C until the culture reached an $OD_{600}$ of ~0.5, at which point IPTG was added to a final concentration of 200 µM and the cell culture temperature was lowered to 15 °C. Following overnight protein expression, the cells were pelleted by centrifugation (4 °C, $5000 \times g$, 10 min) and the supernatant poured off. Cell pellets were subsequently frozen in liquid nitrogen, and stored at −80 °C.

~10 g cell pellets were thawed and resuspended in 50 mL prechilled cell lysis buffer (100 mM Tris-HCl pH 7.5, 100 mM NaCl, 10 mM BME, 5% glycerol) and lysed via sonication. After sonication, the cell solution was moved to a 50 mL conical tube and the cell remains were pelleted by centrifugation (4 °C, 30 min, $10,000 \times g$). The supernatant was moved to a new 50 mL conical tube and again cleared by

centrifugation (4 °C, 30 min, 10,000 × $g$). The supernatant was moved to a new 50 mL conical tube and held on ice. 2 mL of HisPur Ni-NTA Resin per 10 g cell pellet were washed repeatedly with column buffer (100 mM Tris-HCl pH 7.5, 100 mM NaCl, 10 mM BME, 10 mM imidazole) and resuspended to a final volume of 2 mL with column buffer. The resin was then added to the cleared cell supernatant in lysis buffer, and the supernatant solution was subsequently gently mixed at room temperature for 30 min. After 30 min, removed supernatant solution and washed 5x with column buffer (beads were centrifuged for 2 min, 4 °C, 700 × $g$, supernatant removed, and beads resuspended with column buffer). After the final wash, removed the supernatant solution was removed and 10 mL of elution buffer (100 mM Tris-HCl pH 7.5, 100 mM NaCl, 10 mM BME, 500 mM imidazole) was added. The beads were incubated on ice for 10–15 min with gentle periodic resuspension, and then were centrifuged (2 min, 4 °C, 700 × $g$). The supernatant solution containing the eluted protein was set aside, and the elution buffer wash and centrifugation repeated. The eluted protein was then stored at 4 °C overnight or was used immediately in the following step.

The eluted protein was subsequently concentrated using an Amicon 3000 kDa cutoff spin filters via centrifugation (4 °C, 4000 × $g$). Following concentration, the protein was buffer exchanged into buffer II (100 mM Tris-HCl pH 7.5, 100 mM NaCl, 10 mM TCEP-HCl) using a PD-10 desalting column. To the eluted protein fractions, 25 µL of (His)$_6$-tagged TEV-protease (Sigma-Aldrich) was added and the solution incubated overnight at 4 °C.

After overnight incubation, ~1.5 mL of HisPur Ni-NTA Resin per protein elution was washed and resuspended in buffer II. The resin was then added to the TEV-protease cleaved protein solution to remove the TEV-protease as well as the cleaved (His)$_6$-tag. The solution was incubated at 4 °C with periodic, gentle mixing for 30 min, and the beads were then pelleted by centrifugation (4 °C, 4000 × $g$). The supernatant containing the purified, (His)$_6$-tag-cleaved protein was put into a new tube, and the protein concentration determined by nanodrop (assuming an extinction coefficient of $\varepsilon_{Molar} = 46410$ M$^{-1}$cm$^{-1}$ for ΔPTPN1 constructs and $\varepsilon_{Molar} = 50880$ M$^{-1}$cm$^{-1}$ for ΔPTPN2 constructs. The protein solutions were moved into 1 mL aliquots in 1.5 mL Eppendorf tubes, which were flash frozen in liquid nitrogen and stored at −80 °C.

### ΔPTPN1 construct Cu$^{1+}$ binding assay
To evaluate Cu$^{1+}$ binding by the ΔPTPN1 constructs, the purified, (His)$_6$-tag-cleaved protein was thawed on ice and diluted with buffer II to a final concentration of 35 µM. To generate Cu$^{1+}$, we used an adapted method from Padilla-Benavides et al.[83]. Specifically, CuCl$_2$ solutions were prepared at the indicated concentrations in buffer II with 100 mM sodium ascorbate included as well, to fully reduce the Cu$^{2+}$ to Cu$^{1+}$. 500 µL of protein solution was mixed with 500 µL of 350 µM CuCl$_2$ in buffer II with 100 mM sodium ascorbate; the solution was incubated on ice for 1 h. After 1 h, the solution was run over a PD-10 desalting column (held at 4 °C) equilibrated with buffer II. The eluted sample was then used to prepare ICP-MS samples.

### ΔPTPN2 construct Cu$^{1+}$ binding
The purified ΔPTPN2 and ΔPTPN2 C216S proteins were thawed on ice, and then moved to room temperature. To each, either buffer II or 10 protein equivalents of glutathione in buffer II was added. Cu$^{1+}$ (in the form of tetrakis(acetonitrile)copper tetrafluoroborate) was dissolved in neat acetonitrile. 5 protein equivalents of Cu$^{1+}$ were then added to the 6.6 µL of this stock solution was then added to the 660 µL solution of protein (or protein with glutathione) for a final Cu1+ concentration of 5 protein equivalents. This mixture was allowed to incubate at room temperature for 15 min, before being run over a PD-10 desalting column. The protein concentration was then measured by nanodrop, and the samples were immediately used to generate ICP-MS samples.

### ΔPTPN1 $p$-Nitrophenyl Phosphate ($p$NPP) Assay
To assess PTP activity of the ΔPTPN1 construct, 100 nM purified, His-tag cleaved ΔPTPN1 was aliquoted into a 96 well plate and incubated for 15 min with Cu$^{1+}$ solution (generated the same as in the ΔPTPN1 Construct Cu$^{1+}$ Binding Assay) at room temperature. 20 µL of 500 mM $p$NPP (New England Biolabs) was then added to the 190 µL protein mixture (per well) to generate a final concentration of 47.62 mM PNPP per reaction. This mixture was immediately monitored (with a Synergy HTX Multi-Mode Reader plate reader) for the absorbance at 405 nm, corresponding to the chromogenic product. All readings were normalized to the initial measurement for each replicate.

### Statistics and reproducibility
Sample size: For all experiments except RNA-qPCR, experimental sample sizes were pre-determined to be performed in triplicate (or duplicate in the case of KAS-seq) so as to be able to evaluate statistical significance and identify any outlier/anomalous replicates. KAS-seq experiments were performed in duplicate due to the cost-prohibitive nature of the overall experiment. These sample sizes are both sufficient to run advanced analysis (e.g., GSEA on RNA-seq samples), while minimizing false positives in the KAS-seq data (e.g., only two "hits" at 15 min post Cu-supplementation). The RNA-qPCR sample sizes were initially envisioned as two biological replicates at each condition (enough for detection of dose-response and statistical significance in changes), but as the experiment evolved and transcripts of interest were added/removed, the previously chosen transcripts (*EGR1*, *CTR1*, *NAB1*, and *NAB2*) were repeatedly evaluated as well. In the case of RNA-qPCR results, the replicates are enough for both statistical significance and to identify trends (such as dose-response) across the Cu supplementation concentrations. Additionally, more biological replicates were performed in response to reviewer comments during a prior journal submission. No statistical method was predetermined beforehand to define sample sizes.

Data exclusions: 1 out of 2 total biological replicates each of *NAB1* and *NAB2* RNA-qPCR (0–200 µM CuCl$_2$, 60 min incubation) were excluded for being anomalous; the replicate fold-changes were >3 standard deviations above the mean. 1 out of 22 total biological replicates of *CTR1* (30–100 uM CuCl$_2$, 4 h incubation) was excluded for being anomalous; the biological replicate fold-changes were >3 standard deviations above the mean for multiple transcript conditions. The exclusion criteria were not pre-established. Inclusion of the *NAB1*, *NAB2*, and *CTR1* 100 µM CuCl$_2$ data points would not affect any conclusions in the manuscript. Inclusion of the *CTR1* 4 h incubation 30 µM CuCl$_2$ would result in the 4 h, 30 µM CuCl$_2$ *CTR1* fold-change increase being statistically significant but would otherwise not affect any of the conclusions in the manuscript.

Replication: All replicates used to derive our conclusions are reported in the manuscript, and were successful in validating the conclusions stated therein.

### Reporting summary
Further information on research design is available in the Nature Portfolio Reporting Summary linked to this article.

## Data availability
The high throughput sequencing data generated in this study have been deposited in the Gene Expression Omnibus (GEO), accession GSE211339 (A549 KAS-seq), GSE210777 (MDA-MB-468 RNA-seq), and GSE214566 (A549 RNA-seq). DESeq2 differential expression analysis results from the various KAS-seq and RNA-seq experiments are provided as Supplementary Data 1 (A549 KAS-seq), 2 (MDA-MB-468 RNA-seq), and 3 (A549 RNA-seq). Metascape gene ontology (GO) enrichment analysis summaries are provided as Supplementary Data 4 (MDA-MB-468 TM-treated) and 5 (2 h, 30 µM CuCl2 treated A549 RNA-seq). Specifically, raw qPCR data (fold-changes), ICP-MS Cu-stoichiometries, and

pNPP Assay absorbance readings are included in the Source Data file. Raw mass photometry movies are available via Figshare with the following DOI; 10 nM TG https://doi.org/10.6084/m9.figshare.26058631.v1; 50 nM TG https://doi.org/10.6084/m9.figshare.26058622.v1; 10 nM BAM https://doi.org/10.6084/m9.figshare.26058643.v1; 50 nM BAM https://doi.org/10.6084/m9.figshare.26058640.v1; sEGFR with buffer added https://doi.org/10.6084/m9.figshare.26058637.v1; sEGFR with 1 equivalent of CuCl$_2$ added https://doi.org/10.6084/m9.figshare.26058634.v1; sEGFR with 10 equivalents of CuCl$_2$ https://doi.org/10.6084/m9.figshare.26058628.v1; sEGFR with 100 equivalents of CuCl$_2$ added https://doi.org/10.6084/m9.figshare.26058625.v1. GTEx and TCGA RNA-seq data were accessed from, and are available via, https://gtexportal.org/home/ and https://xenabrowser.net/, respectively. ChIP-Atlas ChIP-seq data are available via https://chip-atlas.org/. CREB target gene database data is available via http://signal.salk.edu/creb/index.html. Source data are provided with this paper.

## Code availability

Example code and pseudocode used for next generation sequencing data processing is provided in the Supplementary Materials section.

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

## Acknowledgements

This work was supported by funding from the National Institutes of Health (R01ES030546 and HG006827) to C.H., R35GM143052 to M.Z., K99ES034084 to M.O.R., the Breast Cancer Research Foundation FP049439 to O.I.O., and University of Chicago Yen Postdoctoral Fellowship to M.O.R. We appreciate the kind support from UChicago genomics facility, Mass Spectrometry Facility, and ARC, especially Dr. C. Jin Qin and Dr. Pieter W. Faber for assisting in the experiments. C.H. is an Investigator of the Howard Hughes Medical Institute. The authors are grateful to Prof. Bruce Marc Bissonnette for the providing Cetuximab antibodies as a kind gift. We thank Prof. Wenbin Lin and Yingjie Fan (University of Chicago) for running the ICP-MS samples, and we thank Prof. Eugene Chang for the use of the anaerobic glovebox.

## Author contributions

Conceptualization: M.O.R. and C.H.; methodology: M.O.R., Y.X., M.Z., and C.H.; investigation: M.O.R., Y.X., R.O., C.Y., O.N. P.Z., R.L., T.W., P.W., and O.K.; funding acquisition: M.O.R., L.L., M.Z., and C.H.; supervision: O.I.O., M.Z., and C.H.; Writing—original draft: M.O.R., Y.X., and R.O.; Writing—review & editing: M.O.R., C.Y., L.L., M.Z., and C.H.

## Competing interests

C.H. is a scientific founder and equity holder of Aferna Green, Ellis Bio, AccuraDX, and Accent Therapeutics, and a scientific advisory board member of Aferna Green, Element Biosciences, and Rona Therapeutics. T.W. is an equity holder of AccuraDX. The remaining authors declare no competing interests.
