## [Peer Review File · Nature Communications]

PTPN2 copper-sensing relays copper level fluctuations into EGFR/CREB activation and associated CTR1 transcriptional repressionREVIEWER COMMENTS

Reviewer #1 (Remarks to the Author):

The manuscript by Ross et al describes the role of CREB and PTPN2 in copper homeostasis. The significance of this manuscript is linked to the critical role copper has in physiology and how it has been targeted in cancer physiology. Cuproplasia, the accumulation of copper in tumors, and cuproptosis, copper mediated cell death has been of subjects of major interest recently in the cancer biology. A missing piece in these studies has been what mediates the sensing mechanism. The data for CREB regulation of CTR1 is sound and is a significant advance in our knowledge. The bioinformatic and experimental evidence that CREB is a physiological regulator of copper homeostasis dovetails nicely with previous evidence that cAMP regulation is mediated by copper.

The manuscript also describes the details of experiments that point to regulation of the MAPK pathway by the regulation of the activity of the phosphatase which in turn regulates CREB. This evidence is highlighted as demonstrating PTPN2 as the copper sensor in the pathway and it is suggested that the data demonstrates this is via direct copper binding. But the manuscript does not provide some elements that would help differentiate between a sensor of copper status and a pathway that is regulated by copper. Several key features would be needed to demonstrate sensing such as the kinetics of binding in a competitive environment, the copper affinity relative to the sensory pool of copper, and the localization of the protein. These characteristics define in significant ways the definition of the sensor protein that would allow them to transduce distinct signals versus a "brute force" copper inactivates the phosphatase activity. The abundance and proposed stoichiometry seem very high for this to be a physiological sensor. The current literature has previously linked copper binding to the map kinase MEK1 in the ERK signal transduction. It is not clear that this is not an additional layer of regulation rather than the ultimate sensing molecule. To help solidify this point the authors should responds to the following points:

Major points to be considered:

- 1) The localization in the cytosol seems appropriate for the sensing of copper in the cell. But have the authors attempted any experiments that demonstrate cytosolic copper levels are directly correlated with responsiveness of CREB. Perhaps experiments with cytosolic chemical sensors or competitors?
- 2) The direct binding of copper to PTPN2 reveals 6 coppers bound to the protein, mutation of the catalytic cysteine drops this to 5. Do the authors have any speculation to the locations of these sites and possible ligands? To act as a sensor the kinetics of binding would be important consideration. What is the on-off rate of binding for the sensing site?
- 3) Do the authors have any data for the binding of copper in competitive environments such as the presence of chelators of different strengths? This would allow you to establish the affinity of the different copper for PTPN2?
- 4) Is there any spectroscopic evidence that the copper is bound to the protein specifically with sulfur coordination? What are the combined ligands of these sites all sulfur or other mixed ligands?

Minor points

- 1) Only a single biomarker of copper status has been proposed and that is the CCS levels. Copper deficiency induces stability of CCS while excess copper leads to its turnover, it would be of interest to understand the status of CCS in experimental set-up. Is the PTPN2 signal pre- or post changes in stability of CCS? A sensor should be more responsive to fluctuations in cellular copper than this biomarker.
- 2) Do changes in DMT1 as the copper(II) transporter regulate PTPN2 activities or change CREB activity?

Reviewer #2 (Remarks to the Author):

The manuscript deals with the ability of low level copper supplementation causes Cu¹⁺ binding to—and inactivation of—the protein tyrosine phosphatase (PTP) PTPN2 at the active site cysteine to drive activation of EGFR signal transduction, leading to MAPK/ERK/CREB activation. Though the same authors state that due to the highly homologous nature of PTPN1 and PTPN2, they do not exclude the possibility that PTPN1 can also function in such a role in different cell contexts. In addition, the authors highlight to have established a link between CREB activation and CTR1 transcriptional suppression is reported.

The experimental efforts appear appreciable and the findings reliable. My concerns regard some statements

i. "We assign PTPN2 as the physiological sensor of copper" and, partly, ii. "...we propose CREB activation may be the underlying driver of this CTR1 repression"

Copper is not the only metal ion of d-block transition metal ions able to inhibit protein tyrosine phosphatases (Singh KB, Maret W. The interactions of metal cations and oxyanions with protein tyrosine phosphatase 1B. *Biomaterials*. 2017;30(4):517-52)

It is well known both that copper induces CREB activation, also in a NGF context, (D. La Mendola et al., Metal ion coordination in peptide fragments of neurotrophins: A crucial step for understanding the role and signaling of these proteins in the brain, *Coord Chem Rev*. 435 (2021) 213790) and that the increase of copper concentration involves the decrease of Ctr1 on the surface of cellular membranes, due to the involvement of the transcription factor Sp1 (Dong Yan et al. Effects of Cu(II) and cisplatin on the stability of Specific protein 1 (Sp1)-DNA binding: Insights into the regulation of copper homeostasis and platinum drug transport. *J. Inorg. Biochem*. 2016; 161: 37–39; Yuan S et al., Copper-finger protein of Sp1: the molecular basis of copper sensing., *Metallomics*. 2017;9(8):1169-1175)

My suggestion is to utilize also the literature findings reported in the above cited notes to obtain a more complete picture of the role of Transcription Factors on Ctr1 expression.

Reviewer #3 (Remarks to the Author):

The manuscript from Ross et al. describes the signal transduction mechanism after intracellular copper accumulation. Using KAS-seq technique, the authors first showed that fluctuations in physiologically-relevant copper levels rapidly modulated EGFR/MAPK/ERK signal transduction activation of the transcription factor cAMP response element-binding protein (CREB). By pairing these results with phosphorylation arrays to define copper-stimulated signal transduction pathways, as well as transcriptomic, molecular biological, and biochemical investigations, the authors showed that Cu could bind and inactivate the protein tyrosine phosphatase (PTP) PTPN2 at the active site cysteine to drive activation of EGFR signal transduction, leading to MAPK/ERK/CREB activation. And CREB activity was inversely correlated with CTR1 expression. Overall, it seems that the authors present an interesting study that pinpoints propose that PTPN2 as a physiological copper sensor and defines a regulatory mechanism linking feedback control of copper-stimulated MAPK/ERK/CREB-signaling and CTR1 expression. However, many data in the manuscript do not support this conclusion.

Major Concerns:

- 1) Line 108: The author added 0-200 μM CuCl₂ to treat cells, but in the Figure 1d, the normal group without CuCl₂ treatment does not exist. A normal control group without CuCl₂ is necessary. The same problem also exists in Supplementary Figures 4 and 6.
- 2) In the cell experiment, the highest CuCl₂ applied by the author within the cell was 200 μM (Figure 4a). In fact, under physiological conditions, there is no such high copper concentration environment within the cell. How does the author consider this?
- 3) Line 742: Figure 1d, the effect of Cu²⁺ on CTR1 expression is very weak. The reviewer thinks it's difficult to support the author's conclusion that adding copper can inhibit the expression of CTR1.
- 4) The vertical coordinates of the control group in the manuscript have not been normalized, and the

- author needs to explain how the relevant data processing is carried out (Figure 1d, Figure 4a)
- 5) Line 757: The normal control group in Figure 2d needs to be displayed, and this result also indicates that CREB has a weak regulatory effect on CTR1 transcription.
 - 6) In Line 757: Figure 2d, compared with the control group, there were no significant changes in the CTR1 and MAPK/ERK pathway related genes in the KRASG12X group.
 - 7) In Figure 1b, low Cu and high Cu are set to 10 and 20 μM , respectively. These two concentrations only differ by twice and are both within the physiological concentration range. In Figure 3b, the author changed the treatment concentration to two concentrations: 30 μM and 200 μM . How does the author consider setting Cu's concentration.
 - 8) According to the author's manuscript, the knockdown of PTPN2 can effectively induce EGFR activation. Therefore, EGR1 should be upregulated. Line 757: The knockdown of PTPN2 in Figure 4a actually down-regulated the expression of EGR1, which is completely opposite to the result of adding Cu treatment (Figure 4d). This result contradicts the conclusion of the manuscript. Line 215: "PTPN2 knockdown significantly diminished copper-stimulated EGR1 activation in A549 cells". This result is also problematic because the knockdown of PTPN2 and the effect of Cu treatment on EGR1 expression are consistent.
 - 9) The data in Figure 5 does not fully reflect the author's intended meaning. The uptake of copper does not inhibit PTPN2, but rather relieves the inhibition of EGFP by PTPN2. The inhibitory arrows in this area are not appropriate.
 - 10) Supplementary data 8 needs to display the control group. In addition, the figure shows 2 points, while the figure legend has 3 repetitions.
 - 11) Line 779: The changes in CTR1 are not shown in Figures 1b and 3b.

Reviewer 1

We thank the reviewer for the careful reading and constructive comments. We appreciate the overall positive feedback and have responded to the points raised by the reviewer, as outlined below.

Major points to be considered:

1) The localization in the cytosol seems appropriate for the sensing of copper in the cell. But have the authors attempted any experiments that demonstrate cytosolic copper levels are directly correlated with responsiveness of CREB. Perhaps experiments with cytosolic chemical sensors or competitors?

This is an excellent consideration. We consequently evaluated A549 cells treated with: *i*) the Cu-ionophore GTSM (to increase cytosolic Cu levels); *ii*) the membrane-permeable Cu¹⁺ sensor CS1 to effectively decrease labile intracellular Cu¹⁺ levels via binding competition; or *iii*) DMSO as a negative control. Relative to control, Cu-GTSM significantly increased *EGR1* levels and decreased *CTRI* levels (even more strongly than Cu-treatment alone), while CS1 decreased *EGR1* levels but did not significantly affect *CTRI* levels. These new results corroborate the mechanism of cytosolic Cu-sensing by PTPN2, and thus, that cytosolic copper levels correlate with responsiveness of CREB. We have added the results to page 11 and Fig. 4e. We want to thank the reviewer for this very helpful suggestion.

2) We respond to comment “2” in individual components below.

The direct binding of copper to PTPN2 reveals 6 coppers bound to the protein, mutation of the catalytic cysteine drops this to 5. Do the authors have any speculation to the locations of these sites and possible ligands?

These are presumably adventitious sites involving Cys/Met residues, given the thiophilic nature of Cu¹⁺. We do not currently have any additional insight into the location of these sites, although the mutagenesis data mentioned by the reviewer showed that the active site Cys is key to inhibition under physiological conditions.

To act as a sensor the kinetics of binding would be important consideration. What is the on-off rate of binding for the sensing site?

While we agree that determination of on/off rates for Cu¹⁺ binding to the PTPN2 active site would be informative, for the reasons outlined below, we have technical challenges to obtain these rates in the current system.

1. As Cu^{1+} is a d^{10} ion, there are no sensitive spectroscopic handles (optical, paramagnetic resonance, etc.) with which to readily assess the Cu-binding. Cu^{1+} -S ligation generally produces a weak, nondescript ligand-to-metal charge transfer band at ~ 260 nm, completely overlapping with the much stronger protein A_{280} . Moreover, given that the other Cu^{1+} binding sites likely also involve Cys ligation, we do not expect a unique spectroscopic identifier with which to temporally assess Cu^{1+} binding to the PTPN2 active site.
2. Cu^{1+} in water is thermodynamically unstable and disproportionates into Cu^0 and Cu^{2+} . Additionally, we expect minimal conformational changes in the protein associated with Cu^{1+} binding. As such, methods for detecting small biophysical/energetic changes associated with ligand binding (e.g. SPR) are unsuitable for this system.

3) Do the authors have any data for the binding of copper in competitive environments such as the presence of chelators of different strengths? This would allow you to establish the affinity of the different copper for PTPN2?

To address Δ PTPN2 binding of Cu^{1+} in a competitive environment, we conducted two experiments. First, we assessed the *p*NPP phosphatase activity of Δ PTPN2 (as indicated by *p*NPP product formation, with a characteristic absorbance maximum at 405 nm) incubated with 10 Cu^{1+} equivalents in the presence of Cu^{1+} -chelators DTT or GSH (glutathione) relative to Δ PTPN2 without addition of Cu^{1+} and/or competitor. For reference, Cu^{1+} GSH $K_D \approx 9 \times 10^{-12} \text{ M}^{-1}$ while DTT $K_D \approx 10^{-15} \text{ M}^{-1}$ (PMID: 21258123 and 28294521).

Unfortunately, as shown below, the addition of Cu^{1+} chelators resulted in *stronger* inhibition of Δ PTPN2 than Cu^{1+} addition alone, perhaps via stabilizing metal-binding to the PTP active site, reminiscent of GSH and DTT binding to the low-coordinate Cu^{1+} site of Atox1, for example (PMID: 12686548). This result is consistent with Δ PTPN2 binding Cu^{1+} in the presence of these chelators.

Next, we assessed the Cu^{1+} binding stoichiometry of ΔPTPN2 (wild-type as well as active site C216S mutant) in the presence or absence of 10 protein equivalents of GSH via ICP-MS (after desalting chromatography to remove non-protein bound Cu^{1+} ; new results on page 10). In both instances, the C216S mutant bound less Cu^{1+} than the ΔPTPN2 protein (although the difference was not statistically significantly). However, consistent with the proposal from the aforementioned Cu^{1+} chelator experiment—that the chelator may stabilize Cu^{1+} binding to the protein—addition of GSH actually increased the Cu^{1+} stoichiometry by ~ 1 Cu eq. relative to protein without addition of GSH. Thus, our results support ΔPTPN2 binding of Cu^{1+} even in the presence of the biologically relevant Cu^{1+} chelator GSH.

4) Is there any spectroscopic evidence that the copper is bound to the protein specifically with sulfur coordination? What are the combined ligands of these sites all sulfur or other mixed ligands?

In response to the first question, we do not have spectroscopic evidence; given the multiple apparent adventitious Cu^{1+} -binding sites, the identification of Cys-S ligation by at least one of the sites (via the aforementioned weak ligand-to-metal charge transfer band) may not be particularly insightful. In response to the second question, given the apparent adventitious Cu^{1+} bound to the protein and the fact that spectroscopic techniques like X-ray absorption spectroscopy (XAS) will report on the average of all Cu sites in the sample, we do not expect such experiments would be particularly insightful. Instead, direct structural characterization (e.g. X-ray crystallography) is required for conclusive definition of the Cu^{1+} ligands, we feel such experiments might be beyond the scope of the study. We hope structural characterization could be conducted in the future.

Minor points

1) Only a single biomarker of copper status has been proposed and that is the CCS levels. Copper deficiency induces stability of CCS while excess copper leads to its turnover, it would be of interest to understand the status of CCS in experimental set-up. Is the PTPN2 signal pre- or post changes in stability of CCS? A sensor should be more responsive to fluctuations in cellular copper than this biomarker.

The reviewer is correct that CCS is the only 'strictly Cu-level' status biomarker. Transcriptional expression of the metallothionein *MTIX* has also been used as a biomarker of Cu levels in response to Cu supplementation (ref. 54-56). The underlined point is an important consideration, as *MTIX* expression can be driven by elevated non-Cu intracellular metal levels. In response to the reviewer's inquiry, we now show that increased *MTIX* expression is driven only at the highest concentrations of Cu supplementation ($\geq 100 \mu\text{M}$), whereas EGFR phosphorylation activation and elevated *EGR1* expression (which we have demonstrated are both consequences of PTPN2 inhibition by Cu) were detected at far lower concentrations (20 μM and 10 μM CuCl_2 , respectively). We now discuss this in the paragraph beginning "For PTPN2 to function as a Cu sensor..." on page 10 and present qPCR evaluation of Cu-stimulated changes in *MTIX* expression in new Supplementary Fig. 18).

2) Do changes in DMT1 as the copper(II) transporter regulate PTPN2 activities or change CREB activity?

This is an interesting hypothesis we did not think of. To evaluate this possibility, we conducted: *i*) *DMT1* siRNA knockdown and *ii*) *DMT1* overexpression via plasmid transfection in A549 cells. We subsequently evaluated Cu-stimulated activation of *EGR1* as a proxy for PTPN2 Cu-inhibition. Neither *DMT1* perturbation significantly affected Cu-stimulated *EGR1* expression activation relative to control (Supplementary Fig. 7), as we now note on page 6. Thus, changes in DMT1 levels do not affect Cu-stimulated PTPN2 inhibition.

Reviewer 2

We are grateful for the reviewer's helpful comments and agree that incorporation of many of the references provided by this reviewer is important towards presenting a complete picture of mammalian copper regulation. We have responded to the proposed reference inclusions below.

My concerns regard some statements

i."We assign PTPN2 as the physiological sensor of copper" and, partly, ii. "...we propose CREB activation may be the underlying driver of this CTR1 repression" Copper is not the only metal ion of d-block transition metal ions able to inhibit protein tyrosine phosphatases (Singh KB, Maret W. The interactions of metal cations and oxyanions with protein

tyrosine phosphatase 1B. *Biometals*. 2017;30(4):517-52). It is well known both that copper induces CREB activation, also in a NGF context, (D. La Mendola et al., Metal ion coordination in peptide fragments of neurotrophins: A crucial step for understanding the role and signaling of these proteins in the brain, *Coord Chem Rev*. 435 (2021) 213790)...

We want to thank the reviewer for these references and comments. Our model benefitted from evaluation of the previous literature, including those references provided by the reviewer, which are now featured in the manuscript.

...and that the increase of copper concentration involves the decrease of Ctr1 on the surface of cellular membranes, due to the involvement of the transcription factor Sp1 (Dong Yan et al. Effects of Cu(II) and cisplatin on the stability of Specific protein 1 (Sp1)-DNA binding: Insights into the regulation of copper homeostasis and platinum drug transport. *J. Inorg. Biochem*. 2016; 161: 37–39; Yuan S et al., Copper-finger protein of Sp1: the molecular basis of copper sensing., *Metallomics*. 2017;9(8):1169-1175).

The physiological relevance of *CTRI* transcriptional regulation by Cu-Sp1 inactivation is a contentious issue in the field, as discussed in ref. 15: "...one lab has reported that the human *CTRI* gene is transcriptionally induced in response to low Cu concentrations by the zinc-finger transcription factor Specificity Protein 1 (Sp1). However, the in vivo relevance of these findings is unclear...". Similarly, we do not believe that *CTRI* downregulation via Cu-Sp1 inactivation reflects a specific intracellular response to fluctuations in Cu levels. If the mechanism occurs in vivo, we suspect it could be a consequence of intracellular metal dyshomeostasis.

Sp1 is a ubiquitous transcription factor that regulates genes involved with most biological functions, with over 12,000 genomic binding sites (PMID 28018142). Accordingly, *Sp1* knockout is embryonic-lethal in mice. Cu-Sp1 inactivation would cause global changes in transcriptional expression of thousands of genes, not just the specific downregulation of *CTRI* transcriptional expression. Sp1 may be involved, but we think it might not be the copper-specific regulator.

Reviewer 3

We appreciate the reviewer's thoughtful consideration and suggestions, and the manuscript has improved substantially as a consequence. We believe that many of the "major issues" raised by the reviewer were misunderstandings due to our previous ambiguous wording/data presentation in the original manuscript. These issues have since been remedied in revision as outlined below.

Major Concerns:

1) Line 108: The author added 0-200 μM CuCl_2 to treat cells, but in the Figure 1d, the normal group without CuCl_2 treatment does not exist. A normal control group without CuCl_2 is necessary. The same problem also exists in Supplementary Figures 4 and 6.

This comment highlighted the fact we insufficiently introduced the experimental results in the initial manuscript. The qPCR results presented in the aforementioned figures are reported as fold-change relative to “the normal group without CuCl₂”. This is now explicitly stated following the first mention of RNA-qPCR experiments. The control group fold-change referenced by the reviewer is necessarily “1” for all conditions, by definition.

2) In the cell experiment, the highest CuCl₂ applied by the author within the cell was 200 μM (Figure 4a). In fact, under physiological conditions, there is no such high copper concentration environment within the cell. How does the author consider this?

The reviewer is correct. Cu levels ≥ 100 μM are indeed non-physiological, as we previously noted (Discussion: “*Given the superphysiological copper supplementation levels required to drive significant CTR1 transcriptional repression...*”). We feel:

1. It is “common” practice in signal transduction characterization to supplement superphysiological signaling molecule concentrations to generate stronger responses. For example, Stanoev et al. (ref. 48) applied 200 ng/mL EGF supplementation when assessing EGFR activation; human serum EGF levels are 3 orders of magnitude less than that (600-800 pg/mL, PMID: 3527488).
2. ≥ 100 μM CuCl₂ supplementation have been used previously to evaluate physiological (e.g. PMID: 12501239) and toxicological/environmental Cu exposure responses (e.g. PMID: 10564177 and 9728050).
3. Cancer patient serum Cu concentrations >60 μM have been previously reported (Table 2 in PMID: 3455680), which is not so far from the 100 μM CuCl₂ supplementation required to drive significant *CTR1* repression and *MTIX* activation.

We do want to mention that except for *CTR1* (and the newly included metal-stress response transcript *MTIX*), all Cu-stimulated transcriptional and signaling responses were also significantly activated at CuCl₂ supplementation levels within the physiological range.

3) Line 742: Figure 1d, the effect of Cu²⁺ on *CTR1* expression is very weak. The reviewer thinks it's difficult to support the author's conclusion that adding copper can inhibit the expression of *CTR1*.

We agree that the Cu-stimulated changes in *CTR1* expression are indeed weak, as noted by the reviewer, and as we have now mentioned throughout the manuscript. However, the response is highly statistically significant, and therefore, provides evidence that Cu-supplementation drives a weak decrease in *CTR1* expression. Moreover, our new experiments evaluating Cu-ionophore treatment drove even stronger *CTR1* repression than Cu treatment alone (Fig. 4e), corroborative of the proposed mechanism.

4) The vertical coordinates of the control group in the manuscript have not been normalized, and the author needs to explain how the relevant data processing is carried out (Figure 1d, Figure 4a)

We feel the results are more clearly presented, and the conclusions more readily apparent, by independently scaling the Y-axes for the different transcripts. Particularly considering the wide range of fold-change responses across the various transcripts. We agree with the reviewer's second point, we were insufficiently clear in the original manuscript, and have made the following changes-

1. Explicitly stating how the fold-change values were calculated (as mentioned in Reviewer 3, comment 1)
2. Included the new reference 68 in the methods section which describes the $2^{-\Delta\Delta Ct}$ method of qPCR quantification we used for data processing

5) Line 757: The normal control group in Figure 2d needs to be displayed, and this result also indicates that CREB has a weak regulatory effect on CTR1 transcription.

We are unsure of what the reviewer is referring to regarding a missing control group in Figure 2d; the non-variant sample dataset is labeled and displayed in blue/gray. In agreement with the reviewer's second assertion, we now include language throughout the manuscript that CREB-mediated regulation of *CTR1* expression is indeed weak.

6) In Line 757: Figure 2d, compared with the control group, there were no significant changes in the CTR1 and MAPK/ERK pathway related genes in the KRASG12X group.

The difference between the expression levels of the two groups is statistically significant for all transcripts except *NABI*, as noted in the original version of the manuscript. Differences between the two groups are de-emphasized by the \log_2 scale y-axis (log-scaling is necessary to clearly visualize all transcripts on the same plot).

7) In Figure 1b, low Cu and high Cu are set to 10 and 20 μM , respectively. These two concentrations only differ by twice and are both within the physiological concentration range. In Figure 3b, the author changed the treatment concentration to two concentrations: 30 μM and 200 μM . How does the author consider setting Cu's concentration.

Briefly, CuCl_2 supplementation levels $\leq 30 \mu\text{M}$ were applied to evaluate and validate Cu-driven responses within the physiologically-relevant Cu supplementation range, while CuCl_2 supplementation levels $\geq 100 \mu\text{M}$ were chosen to drive stronger activation of the responses, at the potential risk of being less physiologically relevant.

We initially evaluated 10 and 20 μM CuCl_2 as low and high Cu level supplementation, respectively, to measure Cu-stimulated dose-responses bound by the upper limit of healthy serum Cu levels based on the numbers reported by the widely cited review, ref. 11 (maximum Cu concentration $\sim 22 \mu\text{M}$). We subsequently discovered ref. 25, an article reporting male and female serum Cu concentrations of $19.81 \mu\text{M} \pm 5.06 \mu\text{M}$ and $25.01 \mu\text{M} \pm 7.09 \mu\text{M}$ (mean and standard deviation), respectively. We then used 30 μM as an upper bound for the physiological limit. In summary, 20

μM was used as an upper physiological limit in the earliest-conducted experiments, while 30 μM was used after.

8) We responded to comment “8” in individual components below.

According to the author's manuscript, the knockdown of *PTPN2* can effectively induce EGFR activation. Therefore, EGR1 should be upregulated.

We did not claim that “knockdown of *PTPN2* can effectively induce EGFR activation”. In fact, it was previously shown that *PTPN2* knockdown does not significantly affect basal EGFR activation levels (Fig. 3e, Stanoev et al., ref. 48).

Line 757: The knockdown of *PTPN2* in Figure 4a actually down-regulated the expression of EGR1, which is completely opposite to the result of adding Cu treatment (Figure 4d). This result contradicts the conclusion of the manuscript.

We thank the reviewer for pointing this out – we agree, we did not adequately present/describe these results – and we have now clarified our interpretation of the results. First, after drawing our attention to this figure, we noticed that due to an oversight, the top row of Figure 4a was missing the statistical-significance stars for all samples and featured a minor mistake on the top row Neg Cu treatment label. This has been fixed.

Second, across most *PTP* knockdowns *EGR1* expression levels weakly decreased relative to the negative control siRNA transfection (Fig. 4a). This is almost certainly due to some off-target effect(s) of the negative control siRNA leading to slightly elevated *EGR1* levels. Indeed, the new experiments assessing *DMT1* knockdown (Reviewer 1, Minor Point 2) also detected similarly decreased *EGR1* expression levels relative to negative control siRNA transfection. We now mention this fact in the final sentences of the paragraph beginning “The highly homologous (99% catalytic site...” on pages 9-10.

Line 215: “*PTPN2* knockdown significantly diminished copper-stimulated EGR1 activation in A549 cells”. This result is also problematic because the knockdown of *PTPN2* and the effect of Cu treatment on EGR1 expression are consistent.

We agree that this point requires greater consideration. As shown in ref. 48 and 62, and now explicitly discussed in the main text (see the page 12 Discussion section), receptor tyrosine kinases (including EGFR) are regulated by a network of PTPs, and knockdown of individual regulatory PTPs does not abrogate substrate-stimulated activation. Our interpretation of this result is that after knockdown of *PTPN2*, the other EGFR-regulating PTP(s) is/are susceptible to Cu-inhibition.

9) The data in Figure 5 does not fully reflect the author's intended meaning. The uptake of copper does not inhibit *PTPN2*, but rather relieves the inhibition of EGFP by *PTPN2*. The inhibitory arrows in this area are not appropriate.

Thank you, Figure 5 was indeed ambiguous for what we intended to convey. We have updated the figure to now more clearly emphasize that the top half depicts elevated copper level conditions, while the bottom half depicts decreased copper level conditions.

10) Supplementary data 8 needs to display the control group. In addition, the figure shows 2 points, while the figure legend has 3 repetitions.

Thank you. Similar to comment 1 from this reviewer, we now note in the figure legend of now-Supplementary Figure 9 that the qPCR experiment reflects a fold-change relative to negative control siRNA transfection. As shown in the image below where we made the data points smaller and changed the color of 1 data point to red, 2 of the data points are so close together as to be nearly indistinguishable, but there are/were 3 data points corresponding to 3 biological replicates.

11) Line 779: The changes in *CTR1* are not shown in Figures 1b and 3b.

Changes in *CTR1* expression were detected by RNA-qPCR 2-4 hours after supplementation of $\geq 100 \mu\text{M}$ CuCl_2 (Fig. 1d). Figure 1b depicts changes in transcriptional expression 15 minutes after 10-20 μM CuCl_2 supplementation, so no change in *CTR1* levels was predicted nor was one detected. While the Figure 3b RNA-seq experiments probing transcriptional changes 2 hours after 200 μM CuCl_2 supplementation did not detect significant changes in *CTR1* expression, the same experiment 4 hours post- CuCl_2 supplementation did (Supplementary Figure 5). Significantly decreased *CTR1* levels were likely not detected in the Figure 3b RNA-seq experiments due to the relatively small changes in *CTR1* expression stimulated by elevated Cu levels and the lack of sensitivity in the transcriptome-wide next generation sequencing technique (RNA-seq) relative to RNA-qPCR.

REVIEWER COMMENTS

Reviewer #1 (Remarks to the Author):

The revision satisfactorily addresses the major points raised in the initial review. The work addresses an important topic and advances our knowledge of copper regulation of physiology.

Reviewer #2 (Remarks to the Author):

The revised version of the manuscript does not take into account my previous suggestion. That is "to obtain a more complete picture of the role of Transcription Factors on Ctr1 expression", discussing the involvement of the transcription factor Sp1 (Dong Yan et al. Effects of Cu(II) and cisplatin on the stability of Specific protein 1 (Sp1)-DNA binding: Insights into the regulation of copper homeostasis and platinum drug transport. *J. Inorg. Biochem.* 2016; 161: 37–39; Yuan S et al., Copper-finger protein of Sp1: the molecular basis of copper sensing., *Metallomics.* 2017;9(8):1169-1175) in comparison with their proposal on the role of CREB on Ctr1 level. In their reply, the authors continue to ignore not only what is reported in the above cited works, but also the following more recent contributions.

Targeting the Copper Transport System to Improve Treatment Efficacies of Platinum-Containing Drugs in Cancer Chemotherapy. Kuo MT, Huang YF, Chou CY, Chen HHW. *Pharmaceuticals (Basel)*. 2021;14(6):549. doi: 10.3390/ph14060549

It was demonstrated that by reducing cellular Cu bioavailable levels by Cu chelators, hCtr1 is transcriptionally upregulated by transcription factor Sp1, which binds the promoters of Sp1 and hCtr1. In contrast, elevated Cu poisons Sp1, resulting in suppression ...

p53 inhibits CTR1-mediated cisplatin absorption by suppressing SP1 nuclear translocation in osteosarcoma.

Yong L, Shi Y, Wu HL, Dong QY, Guo J, Hu LS, Wang WH, Guan ZP, Yu BS. *Front Oncol.* 2023;12:1047194. doi: 10.3389/fonc.2022.1047194

It was verified that SP1 is directly bound with CTR1 promoter. SP1 overexpression stimulated CTR1 expression, and SP1 knock-down attenuated CTR1 expression. CONCLUSION: The p53 might function as a negative regulator in CTR1 mediate ...

Consequently, I cannot recommend the manuscript for publication.

Reviewer #3 (Remarks to the Author):

The authors have answered my concerns and revised the manuscript satisfyingly. The quality of this article is greatly improved.

Reviewer 2

The revised version of the manuscript does not take into account my previous suggestion. That is “to obtain a more complete picture of the role of Transcription Factors on Ctr1 expression”, discussing the involvement of the transcription factor Sp1 (Dong Yan et al. Effects of Cu(II) and cisplatin on the stability of Specific protein 1 (Sp1)-DNA binding: Insights into the regulation of copper homeostasis and platinum drug transport. J. Inorg. Biochem. 2016; 161: 37–39; Yuan S et al., Copper-finger protein of Sp1: the molecular basis of copper sensing., Metallomics. 2017;9(8):1169-1175) in comparison with their proposal on the role of CREB on Ctr1 level. In their reply, the authors continue to ignore not only what is reported in the above cited works, but also the following more recent contributions.

Targeting the Copper Transport System to Improve Treatment Efficacies of Platinum-Containing Drugs in Cancer Chemotherapy. Kuo MT, Huang YF, Chou CY, Chen HHW. *Pharmaceuticals* (Basel). 2021;14(6):549. doi: 10.3390/ph14060549
It was demonstrated that by reducing cellular Cu bioavailable levels by Cu chelators, hCtr1 is transcriptionally upregulated by transcription factor Sp1, which binds the promoters of Sp1 and hCtr1. In contrast, elevated Cu poisons Sp1, resulting in suppression ...

p53 inhibits CTR1-mediated cisplatin absorption by suppressing SP1 nuclear translocation in osteosarcoma. Yong L, Shi Y, Wu HL, Dong QY, Guo J, Hu LS, Wang WH, Guan ZP, Yu BS. *Front Oncol.* 2023;12:1047194. doi: 10.3389/fonc.2022.1047194
It was verified that SP1 is directly bound with CTR1 promoter. SP1 overexpression stimulated CTR1 expression, and SP1 knock-down attenuated CTR1 expression. CONCLUSION: The p53 might function as a negative regulator in CTR1 mediate ...

We sincerely thank the reviewer for providing the additional references discussing *CTR1* expression regulation via the transcription factor SP1. As transcriptional regulation of *CTR1* is indeed a component of our story, we have incorporated these references (and others relevant to the SP1/*CTR1* model) into the discussion section of the main text. Specifically, we have revised the discussion section to present *i*) background and references in support of this model, *ii*) conflicting results, and *iii*) a potential reconciliation.

Additionally, we would like to clarify our view on the matter to the reviewer. We agree that there is clear, strong evidence SP1 positively regulates *CTR1* transcriptional expression. As such, decreasing SP1 TF activity (whether by inhibition or decreased SP1 expression levels) decreases *CTR1* expression. However, the model of copper-stimulated SP1 inhibition proposed by Liang et al. (ref. 68) is inconsistent with our results as well as those of others, as now described in the discussion section. Thus, our proposed reconciliation

is that copper-stimulated inhibition of SP1 may be cellular context-dependent. Please see page 13 for our added discussion.

REVIEWERS' COMMENTS

Reviewer #2 (Remarks to the Author):

The authors have answered my concerns and revised the manuscript satisfyingly